# SIMPLE DATA SHARING FOR MULTI-TASKED GOAL-ORIENTED PROBLEMS

## ABSTRACT

Many important sequential decision problems – from robotics, games to logistics – are multi-tasked and goal-oriented. In this work, we frame them as Contextual Goal Oriented (CGO) problems, a goal-reaching special case of the contextual Markov decision process. CGO is a framework for designing multi-task agents that can follow instructions (represented by contexts) to solve goal-oriented tasks. We show that CGO problem can be systematically tackled using datasets that are commonly obtainable: an unsupervised interaction dataset of transitions and a supervised dataset of context-goal pairs. Leveraging the goal-oriented structure of CGO, we propose a simple data sharing technique that can provably solve CGO problems offline under natural assumptions on the datasets' quality. While an offline CGO problem is a special case of offline reinforcement learning (RL) with unlabelled data, running a generic offline RL algorithm here can be overly conservative since the goal-oriented structure of CGO is ignored. In contrast, our approach carefully constructs an augmented Markov Decision Process (MDP) to avoid introducing unnecessary pessimistic bias. In the experiments, we demonstrate our algorithm can learn near-optimal context-conditioned policies in simulated CGO problems, outperforming offline RL baselines.

## 1 INTRODUCTION

Goal-Oriented (GO) problems (Kaelbling, 1993) are an important class of sequential decision-making problems with widespread applications, ranging from robotics (Yu & Mooney, 2023) to game-playing (Hessel et al., 2019) to real-world logistics (Mirowski et al., 2018). Many of these problems are multi-tasked: rather than aiming toward a single goal, the agent is expected to reach some goal in a task-specific goal set based on the task instruction. We frame these multi-tasked goal-oriented applications as Contextual GO (**CGO**) problems and design a simple algorithm that can provably solve them using offline datasets that are commonly available in CGO applications.

CGO problem is a special case of contextual Markov Decision Process (MDP) (Hallak et al., 2015). In a CGO problem, each task is a reaching problem with a goal set that is communicated indirectly to the agent via a context. CGO problem includes the classical GO problem as a special case, where the context is just the target goal, but in general contexts in CGO problem can convey rich, high-level task instructions. In robotics, e.g., common contexts are verbal instructions like "clean up the table" whereas goals are specific configurations (e.g., a clean table) in the environment. In games, contexts can be side-quests for the player to accomplish, and in logistics contexts describe origins and destinations of journeys an operator should execute. We will use navigation as a running example in this paper. Imagine instructing a truck operator with the context "Deliver goods to a warehouse in the Bay area". Given the context, they must first infer some feasible goal in the goal set (e.g., a warehouse location) and implement a policy to efficiently navigate to that goal.

CGO problems are challenging, because the rewards are sparse (non-zero rewards only when reaching goals) and the contexts can be difficult to interpret into feasible goals. However, CGO problem has an important structure that the transition dynamics (e.g., navigating a city road network) are independent of the context (e.g., journey origin and destination), and efficient multitask learning can be achieved by sharing dynamics data across tasks or contexts.

We study offline Reinforcement Learning (RL) for CGO problems. Offline learning is timely for CGO problems given the recent availability of suitable massive datasets. We identify two different

kinds of datasets that are commonly available in CGO applications – an (unsupervised) *dynamics* dataset of agent trajectories, and a (supervised) *context-goal* dataset of pairs of contexts and goals. In robotics, task-agnostic play data can be obtained at scale (Lynch et al., 2020; Walke et al., 2023) in an unsupervised manner whereas instruction datasets (e.g., Misra et al. (2016)) allow supervised learning of the context-goal mapping. In navigation, self-driving car trajectories (e.g., Wilson et al. (2021); Sun et al. (2020)) allow us to learn dynamics whereas landmarks datasets (e.g. Mirowski et al. (2018); Hahn et al. (2021)) allow us to map the contexts to goals.

We propose a Simple Data Sharing (SDS) technique that can provably solve CGO problems subject to natural assumptions on the datasets' quality. We prove that SDS can learn a goal-reaching policy for the CGO problem with high probability, as long as 1) the distribution generating the dynamics dataset "covers" a feasible path to the target goal set and 2) the context-goal distribution "covers" some feasible goals. We remark that "covering" here does not mean that such a path or a goal has to appear in the datasets or the data have to be generated only by goal-reaching policies (e.g., it suffices that data can be stitched together to form a goal-reaching path). Instead, it is a distribution-level coverage requirement; we will present coverage conditions based on the generalization ability of function approximators used in learning, which is weaker than requiring coverage conditions based on non-zero density. SDS is a reduction-based technique that can be implemented on top of a standard offline RL algorithm based on Bellman equation. Our key insight is to carefully construct an action-augmented MDP such that the dynamics dataset and context-goal dataset can be reconciled together as a standard reward-labeled offline dataset.

To our knowledge, SDS is the first offline algorithm that can provably solve CGO problems with just positive data (i.e., the context-goal dataset). While the offline CGO problem here can be cast as an offline RL problem with unlabeled data (i.e., viewing each {context, state} pair as a composite state[1]), existing theoretical results (Yu et al., 2022; Hu et al., 2023; Li et al., 2023a) indicate that both positive data and negative data (i.e., pairs of context and non-goal data) are needed. [2]. An alternative approach to offline CGO problems is to predict goals based on contexts and then run offline goal-conditioned RL (Ma et al., 2022). This approach only needs positive data in learning the predictor, but it can fail when the predicted goal is not reachable from the initial state (since the context-goal dataset could contain non-feasible goals).

We contribute an effective SDS technique and a new analysis technique that formally proves that CGO problem can be solved offline with just dynamics data and context-goal data (i.e. positive data), without the need for negative data. We also show that SDS can be implemented on top of existing offline RL algorithms (with concrete instantiations for PSPI (Xie et al., 2021) in Section 3.3 and IQL (Kostrikov et al., 2021) in Section 4). In addition to theoretical analyses, we conduct several experiments in simulated domains, confirming that SDS outperforms SOTA offline RL baselines designed for unlabeled data and the goal prediction baseline. Finally, we situate our contributions within the vast literature on Goal-Oriented RL (Kaelbling, 1993) and contextual MDPs (Hallak et al., 2015) in Section 1.1 and Appendix A.

## 1.1 RELATED WORK

**Offline RL** Offline RL methods have proven to be effective in GO problems as it also allows learning a common set of sub-goals/skills (Chebotar et al., 2021; Ma et al., 2022; Yang et al., 2023). A variety of approaches are used to mitigate the distribution shift between the collected datasets and the trajectories likely to be generated by learnt policies: (1) constrain target policies to be close to the dataset distribution (Fujimoto et al., 2019; Wu et al., 2019; Fujimoto & Gu, 2021), (2) incorporate value pessimism for low-coverage or Out-Of-Distribution states and actions (Kumar et al., 2020; Yu et al., 2020; Jin et al., 2021) and (3) adversarial training via a two-player game (Xie et al., 2021; Cheng et al., 2022). Our SDS allows the use of generic offline RL algorithms to solve CGO problem offline. We demonstrate its applicability with PSPI (Xie et al., 2021) and IQL (Kostrikov et al., 2021) as our base offline RL algorithm in analyses (Section 3.3) and experiments (Section 4), respectively.

---

[1]Context-goal data can be processed into reward-labeled data, whereas dynamics data from the original MDP imputed with all of the contexts seen in the context-goal dataset becomes the reward-unlabeled data.

[2]Additionally reward-labeled data covering the full trajectory is necessary for general offline RL. But for GO problems, we show that a weaker condition of covering only the goals is sufficient. Existing algorithms for offline RL with unlabeled data may work with this weaker notion of coverage, but it is unclear how to prove it.

(a) Similar goal sets with different contexts
(b) Distinct goal sets with finite contexts
(c) Goal sets (could be overlapping) given continuous contexts

Figure 1: The interplay between contexts and goals in a Contextual Goal-Oriented (CGO) problem characterizes many real-world multi-task settings. (a) All the contexts may share similar goal sets (e.g., pouring coffee). (b) A finite set of context, where each context may map to different goal sets. (c) The goal sets can be defined by continuous contexts, and the goal sets from different contexts could share overlapping support so the context-goal relationship is neither one-to-many or many-to-one, creating a complex CGO problem.

**Offline RL with unlabeled data** Our CGO setting is a special case of offline RL with unlabeled data, or more broadly the offline policy learning from observations paradigm (Li et al., 2023a). There only a subset of the offline data is labeled with rewards (in our setting, that is the contexts dataset, as we don't know which samples in the dynamics dataset are goals.). However, the MAHALO scheme in (Li et al., 2023a) is much more general than necessary for CGO problems, and we show instead that our simple data sharing scheme has better theoretical guarantees than MAHALO in Section 3.3. In our experiments, we compare CGO with several offline RL algorithms designed for unlabeled data: UDS (Yu et al., 2022) where unlabeled data is assigned zero rewards and PDS (Hu et al., 2023) where a pessimistic reward function is estimated from a labeled dataset.

**Goal-oriented RL** GO RL has been extensively studied (Kaelbling, 1993). Existing work focus on two critical aspects of goal-oriented RL: (1) data relabeling and augmentation methods to make better use of available data and (2) learning reusable skills to solve long-horizon problems by chaining sub-goals or skills. For (1), hindsight relabeling methods (Andrychowicz et al., 2017; Li et al., 2020) are effective in improving the learning efficiency of agents by reusing visited states in the trajectories as successful goal examples. For (2), hierarchical methods for determining sub-goals, and training goal reaching policies have been effective in long-horizon problems (Nair & Finn, 2019; Singh et al., 2020; Chebotar et al., 2021). Beyond data efficiency, another key objective of goal-oriented RL is generalization, wherein a common representation of target goals is learned. Popular strategies for goal generalization include universal value function approximators (Schaul et al., 2015), unsupervised representation learning (Nair et al., 2018; Nair & Finn, 2019; Han et al., 2021), and pessimism-induced generalization in offline GO formulations (Yang et al., 2023). Our CGO framing enables both data reuse and goal generalization, by using rich contextual representations of goals and a reduction to offline RL to combine dynamics and context-goal datasets.

## 2 PRELIMINARIES

### 2.1 CONTEXTUAL GOAL-ORIENTED (CGO) PROBLEM

A Contextual Goal-Oriented (CGO) problem describes a multi-tasked goal-oriented setting with a *shared* transition kernel. We consider a Markovian CGO problem with an infinite horizon, defined by the tuple $\mathcal{M} = (\mathcal{S}, \mathcal{A}, P, R, \gamma, \mathcal{C}, d_0)$, where $\mathcal{S}$ is the state space, $\mathcal{A}$ is the action space, $P : \mathcal{S} \times \mathcal{A} \to \Delta(\mathcal{S})$ is the transition kernel, $R : \mathcal{S} \times \mathcal{C} \to \{0, 1\}$ is the reward function, $\gamma \in [0, 1)$ is the discount factor, $\mathcal{C}$ is the context space, and finally $\Delta$ denotes the space of distributions. We do not assume any particular topology on $\mathcal{S}, \mathcal{A}$ and $\mathcal{C}$ and they can be continuous. Each context $c \in \mathcal{C}$ specifies a goal-reaching task with a goal set $G_c \subset \mathcal{S}$, and reaching any goal in the goal set $G_c$ is regarded as successful. The reward function is hence defined as $R(s, c) = \mathbb{1}(s \in G_c)$. An episode of a CGO problem starts from an initial state $s_0$ and a context $c$ sampled according to a distribution $d_0(s_0, c)$, and it terminates when the agent reaches the goal set $G_c$. During the episode, $c$ does not change; only $s_t$ changes (according to $P(s'|s, a)$) and the transition kernel $P(s'|s, a)$ is context independent. The classical GO problem (Kaelbling, 1993) is a special case of CGO, where a multi-goal problem can be viewed as multiple contexts with each context describing a goal.

**Spectrum of CGO Problem** Figure 1 illustrates different CGO problems encountered when learning a language-conditioned control policy for a robot manipulator. $s$ describes the robot and the world state, $a$ is the robot action, and $c$ is the language instruction. For each instruction $c$, the manipulation task for the robot is a reaching problem to a set of targeted robot and world states. The simplest CGO instance is when most of the contexts $c \in \mathcal{C}$ correspond to the very similar goal sets, as shown in Figure 1a. In this case, a context-agnostic policy can be near-optimal[3]. When different contexts have non-overlapping goal sets $G_c$ and we only have finite contexts (as in Figure 1b), the problem is essentially multi-task RL which *requires* context-conditioned policies. In its full complexity, the number of contexts can be infinite; and goal sets of different contexts could potentially overlap[4], as shown in Figure 1c. A CGO agent thus needs to learn how to respond to different contexts as well as transfer knowledge efficiently across contexts.

**Objective** Since the context carries rich information, a CGO policy in general is context-conditioned, i.e., $\pi : \mathcal{S} \times \mathcal{C} \to \Delta(\mathcal{A})$. The performance of a policy $\pi$ is measured by its return, $J(\pi) \coloneqq \mathbb{E}_{\pi,P,d_0}\left[\sum_{=0}^{T} \gamma^t R(s_t, c)\right]$, where $T$ is the time the agent first enters $G_c$ (a random variable dependent on $\pi$, $P$ and $d_0$), and $\mathbb{E}_{\pi,P,d_0}$ denotes the expectation over trajectories generated by running $\pi$ with $P$ starting from $s_0, c$ sampled from $d_0$. We can view the return as the average success rate of reaching *any* goal in the goal set $G_c$ when the problem horizon is exponentially distributed (according to the discount $\gamma$). A CGO algorithm takes a policy class $\Pi = \{\pi : \mathcal{S} \times \mathcal{C} \to \Delta(\mathcal{A})\}$ as input and returns a near-optimal policy $\pi^\dagger$ such that $J(\pi^\dagger) \approx \max_{\pi \in \Pi} J(\pi)$.

## 2.2 OFFLINE LEARNING

We aim to solve CGO problems using offline datasets without additional online environment interactions, à la offline RL. We identify two types of data that are commonly available: $D_{\mathrm{dyn}} \coloneqq \{(s, a, s')\}$ is an *unsupervised* dataset of agent trajectories collected from $P(s'|s, a)$, whereas $D_{\mathrm{goal}} \coloneqq \{(c, s) : s \in G_c\}$ is a *supervised* dataset of context-goal pairs. Different offline CGO algorithms can be judged based on the assumptions they require on $\{D_{\mathrm{dyn}}, D_{\mathrm{goal}}\}$, such as what the datasets should cover and how much data are needed to learn $\pi^\dagger$. No algorithm, to our knowledge, can *provably* learn near-optimal $\pi^\dagger$ using *only* the positive $D_{\mathrm{goal}}$ data (i.e., without needing additional *negative* data of non-goal examples) when combined with $D_{\mathrm{dyn}}$ data. In the next section, we demonstrate how to leverage the special structure of the CGO problem to design provably correct offline algorithms. This insight leads to a Simple Data Sharing (SDS) scheme that can enable existing offline RL algorithms (designed for fully labeled data) to solve offline CGO problems using *just* the positive goal-labeled data without needing any additional non-goal examples, or reward learning.

## 2.3 NOTATION AND ASSUMPTION

Before presenting the main results, we introduce some definitions and shorthand to make the presentation more readable. We introduce a fictitious zero-reward absorbing state $s^+ \notin \mathcal{S}$ and modify the dynamics such that whenever the agent enters $G_c$ it transits to $s^+$ in the next time step (for all actions) and stays there forever. This is a standard technique to convert a goal reaching problem (with a random problem horizon) to an infinite horizon problem. It does *not* change the problem.

Specifically, we extend the reward and the dynamics as follows: We define $\bar{\mathcal{S}} = \mathcal{S} \bigcup \{s^+\}$, $\mathcal{X} \coloneqq \mathcal{S} \times \mathcal{C}$, and $\bar{\mathcal{X}} \coloneqq \bar{\mathcal{S}} \times \mathcal{C}$. In addition, we define $\mathcal{X}^+ \coloneqq \{x : x = (s, c), s = s^+, c \in \mathcal{C}\}$. We use $G$ to denote the goal set on $\mathcal{X}$, i.e., $G \coloneqq \{x \in \mathcal{X} : x = (s, c), s \in G_c\}$. With abuse of notation, we define the reward function and the transition kernel on $\bar{\mathcal{X}}$ accordingly as $R(x) = \mathbb{1}(s \in G_c)$ and $P(x'|x, a) \coloneqq P(s'|s, c, a)\mathbb{1}(c' = c)$, where $P(s'|s, c, a) \coloneqq \mathbb{1}(s' = s^+)$ if $s \in G_c$ or $s = s^+$ otherwise $P(s'|s, c, a) = P(s'|s, a)$, where $x = (s, c)$ and $x' = (s', c')$. Notice the context does not change in the transition. For all value functions, we define their value at $s^+$ as zero.

Given a policy $\pi : \mathcal{X} \to \Delta(\mathcal{A})$, we define its state-action value function (i.e., Q function) as $Q^\pi(x, a) \coloneqq \mathbb{E}_{\pi,P}\left[\sum_{t=0}^{\infty} \gamma^t R(x)|x_0 = x, a_0 = a\right]$. We use $V^\pi(x) \coloneqq Q^\pi(x, \pi)$ to denote the value function given $\pi$, where $Q(x, \pi) \coloneqq \mathbb{E}_{a \sim \pi}[Q(x, a)]$. By construction, we have $Q^\pi(x, a), V^\pi(x) \in$

---

[3]Indeed we show in Section 4 that some existing multi-task RL benchmarks are in this regime where a context-agnostic Implicit Q-Learning (IQL) (Kostrikov et al., 2021) baseline performs well.

[4]Here the intersection of all goal sets given all possible contexts can be empty, which means it cannot be solved by a context agnostic policy.

$[0, 1], \forall x \in \mathcal{X}, a \in \mathcal{A}$. By these definitions, we can write the return $J(\pi) = V^\pi(d_0) = Q^\pi(d_0, \pi)$. We denote $\pi^*$ as the optimal policy and define $Q^* \coloneqq Q^{\pi^*}, V^* \coloneqq V^{\pi^*}$.

**Data Assumption**    We suppose that there are two distributions $\mu_{\text{dyn}}(s, a, s')$ and $\mu_{\text{goal}}(s, c)$, where $\mu_{\text{dyn}}(s'|s, a) = P(s'|s, a)$ and $\mu_{\text{goal}}$ has support within $G_c$, i.e., $\mu_{\text{goal}}(s|c) > 0 \Leftrightarrow s \in G_c$. We assume that $D_{\text{dyn}}$ and $D_{\text{goal}}$ are i.i.d. samples drawn from $\mu_{\text{dyn}}$ and $\mu_{\text{goal}}$, i.e.,

$$D_{\text{dyn}} = \{(s_i, a_i, s'_i) \sim \mu_{\text{dyn}}\} \quad \text{and} \quad D_{\text{goal}} = \{(s_j, c_j) \sim \mu_{\text{goal}}\}.$$

We suppose that $x \sim d_0$ is not in $G$ almost surely. This is to simplify the presentation. If $x \in G$, the agent reaches its goal immediately and no learning is needed.

For CGO problems to be solvable offline, we need some additional assumptions about the support of $\mu_{\text{dyn}}$ and $\mu_{\text{goal}}$. We defer the specific assumptions needed for our algorithm to Section 3.3.

## 3    Simple Data Sharing To Solve CGO Problems

The key idea of SDS is the construction of an *action*-augmented MDP with which the dynamics and context-goal datasets can be combined into a conventional offline RL dataset. In the following, first we describe this action-augmented MDP (Section 3.1) and show that it preserves the optimal policies of the original MDP (Appendix B.1). We then outline a practical algorithm to convert the two datasets of an offline CGO problem into a dataset for this augmented MDP (Section 3.2) such that any generic offline RL algorithm can be used as a solver. Finally, in Section 3.3, we theoretically analyze an instantiation of SDS based on PSPI (Xie et al., 2021) and show that SDS can provably find a near-optimal policy for the CGO problem.

### 3.1    Action-Augmented MDP

One reason why offline RL cannot directly leverage $D_{\text{dyn}}$ and $D_{\text{goal}}$ to solve a CGO problem is that each goal-reaching problem has its own context-specific termination criterion. Notice that although the dynamics datasets $D_{\text{dyn}}$ is consistent with the original MDP transition kernel (i.e. $P(s'|s, a)$), it is however not consistent with the transition kernel $P(x'|x, a)$ (which also includes the effect of context-specific termination) of the context-augmented MDP in Section 2.3. This is easiest to see if some $s \in G_c$ in the $D_{\text{goal}}$ dataset is also observed in the dynamics dataset. $D_{\text{dyn}}$ will imply from $(s, a, s')$ that action $a$ can transition to $s'$, however $D_{\text{goal}}$ implies that all actions at $s$ will transition to $s^+$. This conflict means that combining the two datasets naively leads to an inconsistent algorithm.

We propose a new augmented MDP, which augments the action space of the context-augmented MDP in Section 2.3. Define $\bar{\mathcal{A}} = \mathcal{A} \bigcup \{a^+\}$, with a fictitious action $a^+ \notin \mathcal{A}$ to avoid conflicts across $D_{\text{dyn}}$ and $D_{\text{goal}}$. The reward in this action-augmented MDP is now *action-dependent*, for $x = (s, c) \in \mathcal{X}$, $\bar{R}(x, a) \coloneqq \mathbb{1}(s \in G_c)\mathbb{1}(a = a^+)$, which means the reward given any action in the original action space $\mathcal{A}$ is 0, and there would be a reward 1 only if $a^+$ is taken in the goal set. And the transition upon taking action $a^+$ is defined as $\bar{P}(x'|x, a^+) \coloneqq \mathbb{1}(s' = s^+)$ and $\bar{P}(x'|x, a) \coloneqq P(s'|s, a)\mathbb{1}(c' = c)$ for other actions, which means when taking $a^+$ the agent would always transit to $s^+$, and the transition remains the same as in the original MDP given any normal action..

We denote this action-augmented MDP as $\overline{\mathcal{M}} \coloneqq (\bar{\mathcal{X}}, \bar{\mathcal{A}}, \bar{R}, \bar{P}, \gamma)$. For a policy $\pi : \mathcal{X} \to \Delta(\mathcal{A})$ and a value function $f : \mathcal{X} \times \mathcal{A} \to [0, 1]$ defined in the original MDP, we define their extensions on $\overline{\mathcal{M}}$:

$$\bar{\pi}(a|x) = \begin{cases} \pi(a|x), & x \notin G \\ a^+, & \text{otherwise} \end{cases} \quad \text{and} \quad \bar{f}_g(x, a) = \begin{cases} g(x), & a = a^+ \text{ and } x \notin \mathcal{X}^+ \\ 0, & x \in \mathcal{X}^+ \\ f(x, a), & \text{otherwise.} \end{cases}$$

The extension of $f$ is based on a function $g : \mathcal{X} \to [0, 1]$ which determines the $\bar{f}_g$ value only at $a^+$.

**Regret Equivalence**    We show in Appendix B.1 (see Lemma B.3) that the regret of a policy extended to the augmented MDP is equal to the regret of the policy in the original MDP describing the CGO problem, and any policy defined in the augmented MDP can be converted into that in the original MDP without increasing the regret. Thus, solving the augmented MDP can yield correspondingly optimal policies for the original problem. We next sketch a practical technique to combine $D_{\text{dyn}}$ and $D_{\text{goal}}$ along with the fictitious action labels $a^+$ such that we can solve the action-augmented MDP effectively.

---

**Algorithm 1** Simple Data Sharing (SDS) for CGO

---

**Input**: Dynamics dataset $D_{\mathrm{dyn}}$, context-goal dataset $D_{\mathrm{goal}}$
    **for** each sample $(s, c) \sim D_{\mathrm{goal}}$ **do**
        Create transition[5] $(x, a^+, 1, x^+)$, where $x = (s, c)$ and $x^+ = (s^+, c)$, add it to $\bar{D}_{\mathrm{goal}}$
    **end for**
    **for** each $(s, a, s') \sim D_{\mathrm{dyn}}$ **do**
        **for** each $(\cdot, c) \sim \mathcal{D}_{\mathrm{goal}}$ **do**
            Create transition $(x, a^+, 0, x')$, where $x = (s, c)$ and $x' = (s', c)$, add it to $\bar{D}_{\mathrm{dyn}}$
        **end for**
    **end for**
**Output**: $\bar{D}_{\mathrm{dyn}}$ and $\bar{D}_{\mathrm{goal}}$

---

**Remark 3.1.** *The difference between the action-augmented MDP and the original MDP is that the rewards of the action-augmented MDP are known from the data (due to its shifted definition), whereas the rewards of the original MDP are missing. This difference allows us to directly apply a reduction from offline CGO to offline RL. In contrast, working with the original MDP requires learning additionally a (pessimistic) reward function before offline RL can be applied.*

## 3.2 PRACTICAL ALGORITHM: SIMPLE DATA SHARING

In Algorithm 1 we sketch our Simple Data Sharing (SDS) technique. It takes the two datasets $D_{\mathrm{dyn}}$ and $D_{\mathrm{goal}}$ as input, and produces a single dataset $\bar{D}_{\mathrm{dyn}} \bigcup \bar{D}_{\mathrm{goal}}$ that is suitable for use by any offline RL algorithm based on Bellman equation like CQL (Kumar et al., 2020), IQL (Kostrikov et al., 2021), PSPI (Xie et al., 2021), ATAC (Cheng et al., 2022) etc. Notice that any policy returned by the offline RL algorithm can be executed in the CGO problem by simply masking out the $a^+$ action. We note that in practice Algorithm 1 can be implemented as a pre-processing step in the minibatch sampling of a deep offline RL algorithm (as opposed to computing the full $\bar{D}_{\mathrm{dyn}}$ and $\bar{D}_{\mathrm{goal}}$ once before learning). Empirically, we found that equally balancing the samples $\bar{D}_{\mathrm{dyn}}$ and $\bar{D}_{\mathrm{goal}}$ generates the best result. Below we analyze SDS theoretically by applying SDS to PSPI (Xie et al., 2021); later in Section 4, we apply SDS to IQL (Kostrikov et al., 2021) in simulation experiments.

## 3.3 INFORMATION THEORETIC GUARANTEE: SDS+PSPI

In this section, we show a formal analysis for our reduction approach, when instantiated with PSPI (Xie et al., 2021). Due to space limit, we present the instantiation of SDS+PSPI algorithm in Appendix B.3, and summarize our assumptions and the main theoretical result as follows.

**Assumption 3.2** (Realizability)**.** *We assume for any $\pi \in \Pi$, $Q^\pi \in \mathcal{F}$ and $R \in \mathcal{G}$.*

**Assumption 3.3** (Completeness)**.** *We assume: For any $f \in \mathcal{F}$ and $g \in \mathcal{G}$, $\max(g(x), f(x, \pi)) \in \mathcal{F}$; And for any $f \in \mathcal{F}$, $\pi \in \Pi$, $\mathcal{T}^\pi f(x, a) \in \mathcal{F}$, where $\mathcal{T}^\pi$ is a zero-reward Bellman backup operator with respect to $P(s'|s, a)$: $\mathcal{T}^\pi f(x, a) := \gamma \mathbb{E}_{x' \sim P(s'|s,a) \mathbb{1}(c'=c)}[f(x', \pi)]$.*

**Definition 3.4.** *We define the generalized concentrability coefficients:*

$$\mathfrak{C}_{dyn}(\pi) := \max_{f, f' \in \mathcal{F}} \frac{\|f - \mathcal{T}^\pi f'\|^2_{\rho^\pi_{\notin G}}}{\|f - \mathcal{T}^\pi f'\|^2_{\mu_{dyn}}} \qquad and \qquad \mathfrak{C}_{goal}(\pi) := \max_{g \in \mathcal{G}} \frac{\|g - R\|^2_{\rho^\pi_{\in G}}}{\|g - R\|^2_{\mu_{goal}}} \tag{1}$$

*where $\|h\|^2_\mu := \mathbb{E}_{x \sim \mu}[h(x)^2]$, $\rho^\pi_{\notin G}(x, a) = \mathbb{E}_{\pi, P}\left[\sum_{t=0}^{T-1} \gamma^t \mathbb{1}(x_t = x, a_t = a)\right]$, $\rho^\pi_{\in G}(x) = \mathbb{E}_{\pi, P}\left[\gamma^T \mathbb{1}(x_T = x)\right]$, and $T$ is the first time the agent enters the goal set.*

Concentrability coefficients is a generalization notion of density ratio; *it describes how much the (unnormalized) distribution in the numerator is "covered" by that in the denominator in terms of the generalization ability of function approximators* (Xie et al., 2021). If $\mathfrak{C}_{\mathrm{dyn}}(\pi)$, $\mathfrak{C}_{\mathrm{goal}}(\pi)$ are finite given $\mu_{\mathrm{goal}}, \mu_{\mathrm{dyn}}, \mathcal{F}, \mathcal{G}, \pi$, then we say $\pi$ is covered by the data distributions and conceptually offline RL can learn a policy to be no worse than $\pi$.

---

[5] $s^+$ is implemented as `terminal`=True.

**Theorem 3.5.** *Let $\pi^\dagger$ denote the learned policy of SDS + PSPI with datasets $D_{dyn}$ and $D_{goal}$, using value function classes* [6] $\mathcal{F} = \{\mathcal{X} \times \mathcal{A} \to [0,1]\}$ *and* $\mathcal{G} = \{\mathcal{X} \to [0,1]\}$. *Under Assumption 3.2, 3.3 and 3.4, with probability $1 - \delta$, it holds, for any $\pi \in \Pi$,*

$$J(\pi) - J(\pi^\dagger) \leq \mathfrak{C}_{dyn}(\pi)\sqrt{\epsilon_{dyn}} + \mathfrak{C}_{goal}(\pi)\sqrt{\epsilon_{goal}} \tag{2}$$

*where $\epsilon_{dyn} = O\left(\frac{\log(|\mathcal{F}||\mathcal{G}||\Pi|/\delta)}{|D_{dyn}|}\right)$ and $\epsilon_{goal} = O\left(\frac{\log(|\mathcal{G}|/\delta)}{|D_{goal}|}\right)$ are statistical errors, and $\mathfrak{C}_{dyn}(\pi)$ and $\mathfrak{C}_{goal}(\pi)$ are concentrability coefficients.*

We can interpret the upper bound in Theorem 3.5 as follows: the statistical errors $\epsilon_{\text{dyn}}, \epsilon_{\text{goal}}$ would decrease as we have more data from $\mu_{\text{goal}}$ and $\mu_{\text{dyn}}$; for any compoarator $\pi$ with finite coefficients $\mathfrak{C}_{\text{dyn}}(\pi), \mathfrak{C}_{\text{goal}}(\pi)$, the final regret upper bound would also decrease. Take $\pi = \pi^*$ as an example. For the coefficients $\mathfrak{C}_{\text{dyn}}(\pi), \mathfrak{C}_{\text{goal}}(\pi)$ to be finite, it means 1) the state-action distribution from the dynamics data "covers" the state-action pairs generated by $\pi^*$, which naturally includes stitching; 2) the support of $\mu_{\text{goal}}$ "covers" the goals $\pi^*$ would reach. The coverage is measured based on the generalization ability of $f$ and $g$ as in Definition 3.4; e.g., if $g(x_1)$ and $g(x_2)$ are similar for $x_1 \neq x_2$, then $x_2$ is within the coverage so long as $x_1$ can be generated by $\mu_{\text{goal}}$. It is important to note that Theorem 3.5 simultaneously apply to all $\pi \in \Pi$ not just $\pi^*$. Therefore, so long as the above coverage conditions hold for any goal-reaching policy $\pi$, the learned policy is guaranteed to be goal-reaching too (when the number of samples is large enough so that the statistical errors can be ignored).

As a result, SDS+PSPI can provably learn with only the positive data (i.e., the context-goal dataset) without the need for additional labeling of non-goal samples. Notice that PSPI is one possible way to achieve pessimistic learning in offline RL, and the same intuition could be extended to other offline RL algorithms based on Bellman equation.

**Remark 3.6.** *MAHALO (Li et al., 2023a) is a SOTA offline RL algorithm that can provably learn from unlabeled data. MAHALO can also be implemented on top of PSPI; however, their theoretical result (Theorem D.1) requires a stronger version concentrability, $\max_{g \in \mathcal{G}} \|g-r\|^2_{\rho^\pi_{\notin G}}/\|g-r\|^2_{\mu_{goal}}$, to be small. In other words, it needs additional labeling of non-goal states.*

## 4 EXPERIMENTS

Through experiments we aim to answer the following questions: 1) Does our method work in scenarios of different context-goal relationships shown in Figure 1, under the data assumptions in Section 2.3? 2) Under each setting, is there any empirical benefit from using SDS, compared with offline RL baselines (for unlabeled data) that require pessimistic reward learning or goal prediction?

### 4.1 ENVIRONMENTS AND DATASETS

**Dynamics dataset.** For all experiments, we use the AntMaze-v2 datasets of D4RL (Fu et al., 2020) as dynamics datasets $D_{\text{dyn}}$; we remove the reward and terminal information labels.

**Context-goal dataset.** We construct three levels of context and goal relationships as shown in Figure 1: 1) Figure 1a where multiple contexts define very similar goal sets (Section 4.3); 2) Figure 1b where the number of contexts is finite with distinct goal sets (Section 4.4); 3) Figure 1c where the contexts are continuous and randomly sampled, the goal sets could overlap (Section 4.5). For each environment, we define a context set and *an oracle function* to tell whether a state is within the goal set; this oracle function is only used in data construction and is not accessible to the algorithms tested here. Then given each context, we select states in the dynamics dataset that satisfy the oracle function to construct the goal examples[7]

**Evaluation.** Section 4.3, 4.4 and 4.5 contain results where the training and testing contexts are sampled from the same distribution; in Section 4.5 we also test the algorithms with a different context distribution. For evaluation, we use[8] the oracle function that defines context-goal sets to provide the reward given a certain context in Section 4.4 and 4.5 . The evaluation of each context is done by 100 episodes. We train each algorithm for 5 seeds and report the statistics.

---

[6]We state a more general result for non-finite function classes in the appendix.

[7]Such oracle is never visible to the algorithm but only to construct the dataset.

[8]Exception: in the original AntMaze, we use the D4RL metric, so the results are comparable to the literature.

## 4.2 METHODS

Here we describe the algorithms compared in the experiments. To facilitate a clean comparison of different conceptual approaches to solving offline CGO problems, we use IQL (Kostrikov et al., 2021) as the backbone offline algorithm for all the methods. The same set of hyperparameters in IQL is used in all experiments. In the experiments, we use the $-1/0$ reward notion, which can be shown to be the same as the $0/1$ reward notion in terms of ranking policies under the discounted MDP setting. Please see Appendix C.1 for detailed hyperparameters of all methods.

**SDS+IQL (Ours).** We apply SDS in Algorithm 1 with IQL as the offline RL algorithm to solve the augmented MDP defined in Section 3.1[9]. More specifically, we set $a^+$ to be an extra dimension in the action space but mask out extra dimension for policy output.

**Reward prediction (RP).** For naive reward prediction, we first convert the context-goal set to a dataset with reward 0 for all $(c, s) \sim D_{\text{goal}}$, and then learn a reward function with the dataset. For policy training, we randomly sample $(s, a, s') \sim D_{\text{dyn}}$ and $c \sim D_{\text{goal}}$ and label the transition with the learned reward: if reward prediction of $(c, s')$ is larger then some threshold, we label the transition with $r = 0$ and $\texttt{terminal} = \text{True}$; otherwise we label the transition with $r = -1$ and $\texttt{terminal} = \text{False}$. Then we apply IQL with this labeled dataset.

**PDS.** For PDS (Hu et al., 2023), we follow the similar procedure as RP but learn a *pessimistic* reward function using ensembles. Then we apply similar steps to label the transitions with contexts and apply IQL with this labeled dataset as RP.

**UDS+RP.** On top of RP, we introduce another possible way to learn a reward function while we construct "non-goal" samples in a pessimistic manner: we also sample random $c \sim D_{\text{goal}}$ and $s \sim D_{\text{dyn}}$ and label it with $r = -1$ similar to the spirit of UDS (Yu et al., 2022), then train the reward function with the combined positive and negative dataset. Then we follow the same steps in RP for policy training with the learned reward function.

**Goal prediction.** We learn a conditional generative model as the goal predictor using classifier-free diffusion guidance Ho & Salimans (2022), where the context in the context-goal dataset serves as the condition and the goal states in the context-goal dataset are used to train the generative model. In the meantime, we learn a general goal-conditioned policy with the dynamics-only dataset using HER Andrychowicz et al. (2017)+IQL. Given a test context, the goal predictor first samples the goal under the condition, and then we use the generated goal as the condition for the output policy.

## 4.3 ORIGINAL ANTMAZE

In the original AntMaze, 2D goal locations (contexts) are sampled from a fixed cell in the maze and perturbed with a small noise, generating very similar goal sets. Our training context set is chosen as 2D locations of the states with terminal=True in the D4RL datasets, and the full state is added as the goal example. [10] Test contexts and environmental evaluation follow the original AntMaze.

**SDS matches the performance of the context-agnostic method under the setting of Fig 1a, and achieves better performance than reward learning baselines.** We show the normalized return in each AntMaze environment for all methods in Table 1. Without the need to learn an extra reward function, our method consistently achieves equivalent or better performance in each environment compared to other reward learning baselines. We observe that our method achieves comparable average performance to the context-agnostic method, given that goal sets are all very similar.[11]

**Reward model evaluation for reward learning baselines.** We also visualize the learned reward model from reward learning baselines[12] to show how good they are at predicting the reward, and

---

[9]We choose IQL since it is a popular practical algorithm to solve AntMaze environments

[10]We notice that the number of goal examples in the original AntMaze is too small to learn a good goal predictor, so we omit the goal prediction baseline under this setting. We will present goal prediction results in Section 4.4,4.5 where we construct more goal examples.

[11]Also, we find that umaze is too easy such that even if the goal labeling is bad it still has a relatively high reward (since the maze is too small), so we also omit umaze in other experiments. Li et al. (2023b) show offline RL algorithms can learn good with goal-reaching data even when the rewards are wrong.

[12]We include the details for reward model evaluation in Appendix C.2.

how it is related to the performance. Take "medium-diverse" and "large-diverse" environments as examples (see Figure 2, 3). Also, we observe that PDS+RP is consistently better at separating positive and negative distributions than plain RP and UDS+RP, so we omit to compare with UDS+RP and plain RP in the rest of the experiments. Intuitively, our method does not require reward learning thanks to the construction of the augmented MDP, which avoids the extra errors in reward prediction.

| Env/Method | SDS (Ours) | PDS | RP | UDS+RP | Context-agnostic IQL |
|---|---|---|---|---|---|
| umaze | 94.8±1.3 | 93.0±1.3 | 50.5±2.1 | 54.3±6.3 | 97.7±1.0 |
| umaze diverse | 72.8±7.7 | 50.6±7.8 | 72.8±2.6 | 71.5±4.3 | 65.5±10.5 |
| medium play | 75.8±1.9 | 66.8±4.9 | 0.5±0.3 | 0.3±0.3 | 75.2±3.4 |
| medium diverse | 84.5±5.2 | 22.8±2.4 | 0.5±0.5 | 0.8±0.5 | 76.0±3.7 |
| large play | 60.0±7.6 | 39.6±4.9 | 0±0 | 0±0 | 45.8±2.6 |
| large diverse | 36.8±6.9 | 30.0±5.3 | 0±0 | 0±0 | 46.7±5.4 |
| average | **70.8** | 50.5 | 20.7 | 21.2 | 67.8 |

Table 1: Normalized return in AntMaze-v2, averaged over 5 random seeds with standard errors.

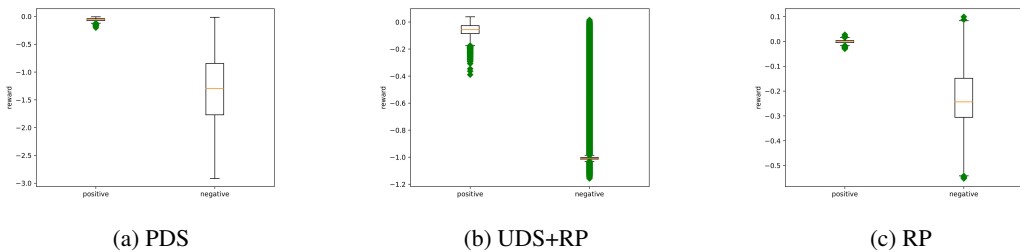

|              (a) PDS                    (b) UDS+RP                    (c) RP |

Figure 2: Reward model evaluation for the large-diverse environment. Green dots are outliers.

## 4.4 MODIFIED ANTMAZE: FOUR ROOMS

**Context-goal setup.** We partition the maze into four rooms and any state in the room would be a goal state. We use discrete room numbers (1,2,3,4) as contexts. As the agent always starts in Room 1, the training and test context sets are Room 2,3,4. We use medium and large environments. Additionally, we perturb all the state dimensions with $\mathcal{N}(0, 0.1)$ Gaussian noises when constructing the context-goal dataset, so the goal sets are not subsets of the dynamics datasets.

**SDS achieves better performance than baselines under the setting in Figure 1b.** We show the normalized return (average success rate in percentage) in each modified Four Rooms environment for our method and baseline methods in Table 2, where our method consistently outperforms the performances of baseline methods. We also provide evaluation for reward learning in Figure 5.

| Env/Method | SDS (Ours) | PDS | Goal Prediction |
|---|---|---|---|
| medium-play | **78.7±0.9** | 46.0±4.47 | 59.3±2.6 |
| medium-diverse | **83.6±1.9** | 51.3±3.6 | 66.7±2.4 |
| large-play | **65.5±2.5** | 13.9±2.4 | 41.4±3.6 |
| large-diverse | **72.2±2.9** | 11.1±3.8 | 42.0±3.0 |
| average | **75.0** | 30.6 | 52.4 |

Table 2: Average scores with standard errors over 5 random seeds from Four Rooms with perturbation. The score for each run is the average success rate (%) of the other three rooms.

## 4.5 MODIFIED ANTMAZE: RANDOM CELLS

**Context-goal setup.** We use the 2D locations as context but the distribution of the context is much more diverse than Section 4.3. For each maze map, we choose a set of non-wall 2D locations in the maze map, uniformly sample from it, and add uniform perturbations to get the training contexts. To construct the goal set given context, we obtain states with the 2D locations within the $L_2$ ball with a certain radius. For test distributions, we have two settings: 1) the same as the training distribution;

| Env/Method | SDS (Ours) | PDS | Goal Prediction |
|---|---|---|---|
| medium-play | **76.8±6.1** | 52.0±8.8 | 66.7±7.2 |
| medium-diverse | **78.2±6.5** | 60.9±11.3 | 69.7±8.7 |
| large-play | **57.6±12.4** | 50.6±6.4 | 42.4±8.2 |
| large-diverse | 54.7±8.8 | **58.3±9.2** | 44.2±8.1 |
| average | **66.8** | 55.5 | 55.8 |

Table 3: Average scores with standard errors over 5 random seeds from Random Rells with perturbation. The score for each run is the average success rate (%) of 5 random test contexts from the same training distribution.

| Env/Method | SDS (Ours) | PDS | Goal Prediction |
|---|---|---|---|
| medium-play | 67.9±8.2 | 50.1±13.4 | **70.5±1.9** |
| medium-diverse | **72.5±6.5** | 57.5±14.8 | 63.0±7.2 |
| large-play | **60.2±4.8** | 48.1±8.0 | 44.3±4.1 |
| large-diverse | **58.0±5.8** | 44.1±9.9 | 55.4±5.7 |
| average | **64.7** | 49.9 | 58.3 |

Table 4: Average scores with standard errors over 5 random seeds from Random Cells with perturbation. The score for each run is the average success rate (%) of 5 random test contexts far away from the start.

2) test contexts are drawn from a limited area that is far away from the starting point of the agent. Additionally, we perturb all the state dimensions with $\mathcal{N}(0, 0.1)$ Gaussian noises when constructing the context-goal dataset, so the goal sets are not subsets of the dynamics datasets.

**SDS outperforms or matches the performance of baselines under the setting in Figure 1c.** We show the normalized return (average success rate in percentage) in each modified Random Cells environment for all methods in Table 3, where our method consistently outperforms all baseline methods in each environment, which also shows the generalization ability in the context space. We also provide reward visualization for reward learning baselines in Figure 6.

**SDS also works with a different test context distribution.** We also test with a different distribution of random cells that are far away from the start with some specified threshold in each environment in Table 4. We can observe that when tested with this different context distribution, SDS still consistently outperforms reward learning baselines.

## 5 CONCLUSION AND LIMITATION

We propose a Simple Data Sharing technique for offline CGO problems. We prove SDS can learn near optimal policies so long as the offline data cover goal-reaching trajectories needed at the test time, without the need of negative labels. We also validate the efficacy of SDS experimentally, and we find it outperforms other reward-learning offline RL baselines across various CGO problem settings. We highlight SDS works under certain assumptions. As shown in our theoretical result in Section 3.3, the SDS technique would fail 1) if the dynamics dataset does not contain trajectories leading to the goal set of a given context, 2) the context-goal dataset does not cover the contexts and goals faced at test time, or 3) if the goal set does not cover reachable goals from initial states. While we believe SDS for its simplicity and theoretical guarantees would be useful in real-world settings (such as learning visual-language robot policies), our experimental setup is limited to low-dimensional simulation environments. Scaling up SDS empirically is an interesting future direction.

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

## A    ADDITIONAL RELATED WORK

**Data-sharing in RL**    Sharing information across multiple tasks is a promising approach to accelerate learning and to identify transferable features across tasks. In RL, both multi-task and transfer learning settings have been studied under varying assumption on the shared properties and structures of different tasks (Zhu et al., 2023; Teh et al., 2017; Barreto et al., 2017; D'Eramo et al., 2020). For data sharing in CGO, we adopt the contextual MDP formulation (Hallak et al., 2015; Sodhani et al., 2021), which enables knowledge transfer via high-level contextual cues. Prior work on offline RL has also shown the utility of sharing data across tasks: hindsight relabeling and manual skill grouping (Kalashnikov et al., 2021), inverse RL (Li et al., 2020), sharing Q-value estimates (Yu et al., 2021; Singh et al., 2020) and reward labeling (Yu et al., 2022; Hu et al., 2023).

## B    SDS +PSPI: THEORETICAL ANALYSIS

In this section, we provide a detailed analysis for the instantiation of SDS using PSPI. We follow the same notation for the value functions, augmented MDP and extended function classes as stated in Section 2 and Section 3 in the main text.

### B.1    EQUIVALENCE RELATIONS BETWEEN ORIGINAL AND AUGMENTED MDP

We begin by showing that the optimal policy and any value function in the augmented MDP can be expressed using their analogue in the original MDP. With the augmented MDP defined as $\overline{\mathcal{M}} := (\bar{\mathcal{X}}, \bar{\mathcal{A}}, \bar{R}, \bar{P}, \gamma)$ in Section 3.1, we first define the value function in the augmented MDP. For a policy $\bar{\pi} : \bar{\mathcal{X}} \to \bar{\mathcal{A}}$, we define the Q function for the augmented MDP as

$$\bar{Q}^{\bar{\pi}}(x, a) := \mathbb{E}_{\bar{\pi}, \bar{P}} \left[ \sum_{t=0}^{\infty} \gamma^t \bar{R}(x, a) | x_0 = x, a_0 = a \right]$$

Notice that we don't have a reaching time random variable $T$ in this definition; instead the agent would enter an absorbing state $s^+$ after taking $a^+$ in the augmented MDP. We can define similarly $\bar{V}^{\bar{\pi}}(s) := \bar{Q}^{\bar{\pi}}(x, \bar{\pi})$.

**Remark B.1.** *Let $\bar{Q}_R^{\pi}$ be the extension of $Q^{\pi}$ based on R. We have, for $x \notin G$, $\bar{Q}_R^{\pi}(x, a) = \bar{Q}^{\bar{\pi}}(x, a)$ $\forall a \in \bar{\mathcal{A}}$, and for $x \in G$, $\bar{Q}_R^{\pi}(x, a) = \bar{Q}^{\bar{\pi}}(x, a^+) = 1$, $\forall a \in \bar{\mathcal{A}}$.*

By the construction of the augmented MDP, it is obvious that the following is true.

**Lemma B.2.** *Given $\pi : \mathcal{X} \to \Delta(\mathcal{A})$, let $\bar{\pi}$ be its extension. For any $h : \mathcal{X} \times \mathcal{A} \to \mathbb{R}$, it holds*

$$\mathbb{E}_{\pi, P} \left[ \sum_{t=0}^{T} \gamma^t h(x, a) \right] = \mathbb{E}_{\bar{\pi}, \bar{P}} \left[ \sum_{t=0}^{\infty} \gamma^t \tilde{h}^{\pi}(x, a) | x \notin \mathcal{X}^+ \right]$$

*where $T$ is the goal-reaching time (random variable) and we define $\tilde{h}^{\pi}(x, a^+) = h(x, \pi)$.*

We can now relate the value functions between the two MDPs.

**Proposition B.3.** *For a policy $\pi : \mathcal{S} \to \Delta(\mathcal{A})$, let $\bar{\pi}$ be its extension (defined above). We have for all $x \in \mathcal{X}$, $a \in \mathcal{A}$,*

$$Q^{\pi}(x, a) \geq \bar{Q}^{\bar{\pi}}(x, a)$$
$$V^{\pi}(x) = \bar{V}^{\bar{\pi}}(x)$$

*Conversely, for a policy $\xi : \bar{\mathcal{X}} \to \Delta(\bar{\mathcal{A}})$, define its restriction $\underline{\xi}$ on $\mathcal{X}$ and $\mathcal{A}$ by translating probability of $\xi$ originally on $a^+$ to be uniform over $\mathcal{A}$. Then we have for all $s \in \mathcal{S}$, $a \in \mathcal{A}$*

$$Q^{\underline{\xi}}(x, a) \geq \bar{Q}^{\xi}(x, a)$$
$$V^{\underline{\xi}}(x) \geq \bar{V}^{\xi}(x)$$

*Proof.* The first direction follows from Lemma B.2. For the latter, whenever $\xi$ takes $a^+$ at some $x \notin G$, it has $\bar{V}^{\xi}(x) = 0$ but $\bar{V}^{\underline{\xi}}(x) \geq 0$ since there is no negative reward in the original MDP. By performing a telescoping argument, we can derive the second claim. $\square$

By this lemma, we know the extension of $\pi^*$ (i.e., $\bar{\pi}^*$) is also optimal to the augmented MDP and $V^*(x) = \bar{V}^*(x)$ for $x \in \mathcal{X}$. Furthermore, we have a reduction that we can solve for the optimal policy in the original MDP by the solving augmented MDP, since

$$V^{\underline{\xi}}(d_0) - V^*(d_0) \leq V^{\xi}(d_0) - \bar{V}^*(d_0)$$

for all $\xi : \bar{\mathcal{X}} \to \Delta(\mathcal{A})$. In particular,

$$\text{Regret}(\pi) := V^{\pi}(d_0) - V^*(d_0) = V^{\bar{\pi}}(d_0) - \bar{V}^*(d_0) =: \overline{\text{Regret}}(\bar{\pi}) \tag{3}$$

Since the augmented MDP replaces the random reaching time construction with an absorbing-state version, the Q function $\bar{Q}^{\bar{\pi}}$ of the extended policy $\bar{\pi}$ satisfies the Bellman equation

$$\bar{Q}^{\bar{\pi}}(x,a) = \bar{R}(x,a) + \gamma \mathbb{E}_{x' \sim \bar{P}(\cdot|x,a)}[\bar{Q}^{\pi}(x',\bar{\pi})]$$
$$=: \bar{\mathcal{T}}^{\pi} \bar{Q}^{\pi}(x,a) \tag{4}$$

For $x \in \mathcal{X}$ and $a \in \mathcal{A}$, we show how the above equation can be rewritten in $Q^{\pi}$ and $R$.

**Proposition B.4.** *For $x \in \mathcal{X}$ and $a \in \mathcal{A}$,*

$$\bar{Q}^{\bar{\pi}}(x,a) = 0 + \gamma \mathbb{E}_{x' \sim \bar{P}(\cdot|x,a)}[\max(R(x'), Q^{\pi}(x',\pi))]$$

*For $a = a^+$, $\bar{Q}^{\bar{\pi}}(x,a^+) = \bar{R}(x,a^+) = R(x)$. For $x \in \mathcal{X}^+$, $\bar{Q}^{\bar{\pi}}(x,a) = 0$.*

*Proof.* The proof follows from Lemma B.5 and the definition of $\bar{P}$. $\qquad\square$

**Lemma B.5.** *For $x \in \mathcal{X}$, $\bar{Q}^{\bar{\pi}}(x,\bar{\pi}) = \max(R(x), Q^{\pi}(x,\pi))$*

*Proof.* For $x \in \mathcal{X}$,

$$
\begin{aligned}
\bar{Q}^{\bar{\pi}}(x,\bar{\pi}) &= \begin{cases} \bar{Q}^{\bar{\pi}}(x,a^+), & \text{if } x \in G \\ \bar{Q}^{\bar{\pi}}(x,\pi), & \text{otherwise} \end{cases} && \text{(Because of definition of } \bar{\pi}) \\
&= \begin{cases} \bar{Q}^{\bar{\pi}}(x,a^+), & \text{if } x \in G \\ Q^{\pi}(x,\pi), & \text{otherwise} \end{cases} && \text{(Because of Proposition B.3)} \\
&= \begin{cases} \bar{R}(x,a^+), & \text{if } x \in G \\ Q^{\pi}(x,\pi), & \text{otherwise} \end{cases} && \text{(Definition of augmented MDP)} \\
&= \begin{cases} R(x), & \text{if } x \in G \\ Q^{\pi}(x,\pi), & \text{otherwise} \end{cases} \\
&= \max(R(x), Q^{\pi}(x,\pi))
\end{aligned}
$$

where in the last step we use $\bar{R}(x) = 1$ for $x \in G$ and $\bar{R}(x) = 0$ otherwise. $\qquad\square$

## B.2 FUNCTION APPROXIMATOR ASSUMPTIONS

In Theorem 3.5, we assume access to a policy class $\Pi = \{\pi : \mathcal{X} \to \Delta(\mathcal{A})\}$. We also assume access to a function class $\mathcal{F} = \{f : \mathcal{X} \times \mathcal{A} \to [0,1]\}$ and a function class $\mathcal{G} = \{g : \mathcal{X} \to [0,1]\}$. We can think of them as approximator for the Q function and the reward function of the original MDP.

Recall the zero-reward Bellman backup operator $\mathcal{T}^{\pi}$ with respect to $P(s'|s,a)$ as defined in Assumption 3.3:

$$\mathcal{T}^{\pi} f(x,a) := \gamma \mathbb{E}_{x' \sim P_0(\cdot|x,a)}[f(x',\pi)]$$

where $P_0(x'|x,a) := P(s'|s,a)\mathbb{1}(c' = c)$. Note this definition is different from the one with absorbing state $s^+$ in Section 2.3. Using this modified backup operator, we can show that the following realizability assumption is true for the augmented MDP:

**Proposition B.6** (Realizability). *By Assumption 3.2 and Assumption 3.3, there is $f \in \mathcal{F}$ and $g \in \mathcal{G}$ such that $\bar{Q}^{\bar{\pi}} = \bar{f}_g$.*

*Proof.* By Assumption 3.3, there is $h \in \mathcal{F}$ such that $h(x, a) = \max(R(x), Q^\pi(x, a))$. By Proposition B.4, we have for $x \in \mathcal{X}$, $a \neq a^+$

$$\bar{Q}^{\bar{\pi}}(x, a) = 0 + \gamma \mathbb{E}_{x' \sim \bar{P}(\cdot|x,a)}[\max(R(x'), Q^\pi(x', \pi))]$$
$$= 0 + \gamma \mathbb{E}_{x' \sim P_0(\cdot|x,a)}[h(x, \pi)]$$
$$= \mathcal{T}^\pi h \in \mathcal{F}$$

For $a = a^*$, we have $\bar{Q}^{\bar{\pi}}(x, a^*) = \bar{R}(x, a^+) = R(x) \in \mathcal{G}$. Finally $\bar{Q}^{\bar{\pi}}(x^+, a) = 0$ for $x^+ \in \mathcal{X}^+$. Therefore, $\bar{Q}^{\bar{\pi}} = \bar{f}_g$ for some $f \in \mathcal{F}$ and $g \in \mathcal{G}$. $\qquad\square$

### B.3 ALGORITHM

In this section, we describe the instantiation of PSPI with SDS in detail along with the necessary notation. As discussed in Section 3.3, our algorithm is based on the idea of reduction, which turns the offline CGO problem into an standard offline RL problem in the augmented MDP. To this end, we construct augmented datasets $\bar{D}_{\text{dyn}}$ and $\bar{D}_{\text{goal}}$ in Algorithm 1 as follows:

$$\bar{D}_{\text{dyn}} = \{(x_n, a_n, r_n, x'_n)|r_n = 0, x_n = (s_i, c_j), x'_n = (s'_i, c_j), a_n = a_i, (s_i, a_i, s'_i) \in D_{\text{dyn}}, (\cdot, c_j) \in D_{\text{goal}}\}$$
$$\bar{D}_{\text{goal}} = \{(x_n, a^+, r_n, x_n^+)|r_n = 1, x_n = (s_n, c_n), x_n^+ = (s^+, c_n), (s_n, c_n) \in D_{\text{goal}}\}$$

For the analysis, we consider a simplified version of Algorithm 1 where we do not reuse the samples in $D_{\text{dyn}}$. Specifically, for each sample $(s_i, a_i, s'_i) \in D_{\text{dyn}}$, we pair it with one sample $(\cdot, c_j) \in D_{\text{goal}}$ and do not reuse the sample from $D_{\text{dyn}}$. This can be naively done by pairing observed transitions and context-goal pairs in both datasets when $|D_{\text{goal}}| \geq |D_{\text{dyn}}|$. In the analysis, we will state our results under this simplification.

With this construction, we have: $\bar{D}_{\text{dyn}} \sim \mu_{\text{dyn}}(s, a, s')\mu_{\text{goal}}(c)$ and $\bar{D}_{\text{goal}} \sim \mu_{\text{goal}}(c, s)\mathbb{1}(a = a^+)\mathbb{1}(s' = s^+)$. With abuse of notation, we write $\mu_{\text{dyn}}(x, a, x') = \mu_{\text{dyn}}(s, a, s')\mu_{\text{goal}}(c)$ and $\mu_{\text{goal}}(x, a, x') = \mu_{\text{goal}}(c, s)\mathbb{1}(a = a^+)\mathbb{1}(s' = s^+)$. Note that, $|\bar{D}_{\text{goal}}| = |D_{\text{goal}}|$ and $|\bar{D}_{\text{dyn}}| = |D_{\text{dyn}}|$ as we are simply augmenting the observed states and actions without reusing samples. These two datasets have the standard tuple format, so we can run offline RL on $\bar{D}_{\text{dyn}} \bigcup \bar{D}_{\text{goal}}$.

**SDS +PSPI**  We consider the information theoretic version of PSPI (Xie et al., 2021) which can be summarized as follows: For an MDP $(\mathcal{X}, \mathcal{A}, R, P, \gamma)$, given a tuple dataset $D = \{(x, a, r, x')\}$, a policy class $\Pi$, and a value class $\mathcal{F}$, it finds the policy through solving the two-player game:

$$\max_{\pi \in \Pi} \min_{f \in \mathcal{F}} \quad f(d_0, \pi) \qquad \text{s.t.} \qquad \ell(f, f; \pi, D) - \min_{f' \in \mathcal{F}} \ell(f', f; \pi, D) \leq \epsilon_b \qquad (5)$$

where $f(d_0, \pi) = \mathbb{E}_{x_0 \sim d_0}[f(x_0, \pi)]$, $\ell(f, f'; \pi, D) := \frac{1}{|D|}\sum_{(x, a, r, x') \in D}(f(x, a) - r - f'(x', \pi))^2$. The term $\ell(f, f; \pi, D) - \min_{f'} \ell(f', f; \pi, D)$ in the constraint is an empirical estimation of the Bellman error on $f$ with respect to $\pi$ on the data distribution $\mu$, i.e. $\mathbb{E}_{x, a \sim \mu}[(f(x, a) - \mathcal{T}^\pi f(x, a))^2]$. It constrains the Bellman error to be small, since $\mathbb{E}_{x, a \sim \mu}[(Q^\pi(x, a) - \mathcal{T}^\pi Q^\pi(x, a))^2] = 0$.

Below we show how to run PSPI to solve the augmented MDP with offline dataset $\bar{D}_{\text{dyn}} \bigcup \bar{D}_{\text{goal}}$. To this end, we extend the policy class from $\Pi$ to $\bar{\Pi}$, and the value class from $\mathcal{F}$ to $\bar{\mathcal{F}}_\mathcal{G}$ using the function class $\mathcal{G}$ based on the extensions defined in Section 3.1. One natural attempt is to implement equation 5 with the extended policy and value classes $\bar{\Pi}$ and $\bar{\mathcal{F}}$ and $\bar{D} = \bar{D}_{\text{dyn}} \bigcup \bar{D}_{\text{goal}}$. This would lead to the two player game:

$$\max_{\bar{\pi} \in \bar{\Pi}} \min_{\bar{f}_g \in \bar{\mathcal{F}}_\mathcal{G}} \quad \bar{f}_g(d_0, \bar{\pi}) \qquad \text{s.t.} \qquad \ell(\bar{f}_g, \bar{f}_g; \bar{\pi}, \bar{D}) - \min_{\bar{f}'_{g'} \in \bar{\mathcal{F}}_\mathcal{G}} \ell(\bar{f}'_{g'}, \bar{f}_g; \bar{\pi}, \bar{D}) \leq \epsilon_b \qquad (6)$$

However, equation 6 is not a well defined algorithm, because its usage of the extended policy $\bar{\pi}$ in the constraint requires knowledge of $G$, which is unknown to the agent.

Fortunately, we show that equation 6 can be slightly modified so that the implementation does not actually require knowing $G$. Here we use a property (Proposition B.4) that the Bellman equation of the augmented MDP:

$$\bar{Q}^{\bar{\pi}}(x, a) = \bar{R}(x, a) + \gamma \mathbb{E}_{x' \sim \bar{P}(\cdot|x,a)}[\bar{Q}^\pi(x', \bar{\pi})]$$
$$= 0 + \gamma \mathbb{E}_{x' \sim \bar{P}(\cdot|x,a)}[\max(R(x'), Q^\pi(x', \pi))]$$

for $x \in \mathcal{X}$ and $a \neq a^+$, and $\bar{Q}^{\bar{\pi}}(x, a) = 1$ for $x \in G$ and $a = a^+$.

We apply these two equalities to $\bar{D}_{\text{dyn}}$ and $\bar{D}_{\text{goal}}$ to construct our Bellman error estimates. Let $\phi(\bar{Q}^{\bar{\pi}}(x)) := \max(R(x), Q^{\pi}(x, \pi))$. We can rewrite the squared Bellman error on these two data distributions using the Bellman backup defined on the augmented MDP (see eq.4) as below:

$$\mathbb{E}_{x,a \sim \mu_{\text{dyn}}}[(\bar{Q}^{\bar{\pi}}(x,a) - \bar{\mathcal{T}}^{\bar{\pi}}\bar{Q}^{\bar{\pi}}(x,a))^2] = \mathbb{E}_{x,a \sim \mu_{\text{dyn}}}[(\bar{Q}^{\bar{\pi}}(x,a) - 0 - \gamma\mathbb{E}_{x' \sim \bar{P}(\cdot|x,a)}[\phi(\bar{Q}^{\bar{\pi}})(x',\pi)])^2] \tag{7}$$

$$\mathbb{E}_{x,a \sim \mu_{\text{goal}}}[(\bar{Q}^{\bar{\pi}}(x,a) - \bar{\mathcal{T}}^{\bar{\pi}}\bar{Q}^{\bar{\pi}}(x,a))^2] = \mathbb{E}_{x,a \sim \mu_{\text{goal}}}[(\bar{Q}^{\bar{\pi}}(x,a^+) - 1)^2] \tag{8}$$

We can construct an approximator $\bar{f}_g(x, a)$ for $\bar{Q}^{\bar{\pi}}(x, a)$. Substituting the estimator $\bar{f}_g(x, a)$ for $\bar{Q}^{\bar{\pi}}(x, a)$ in the squared Bellman errors above and approximating them by finite samples, we derive the empirical losses below.

$$\ell_{\text{dyn}}(\bar{f}_g, \bar{f}'_{g'}; \bar{\pi}) := \frac{1}{|\bar{D}_{\text{dyn}}|} \sum_{(x,a,r,x') \in \bar{D}_{\text{dyn}}} (f(x,a) - \gamma\max(g'(x'), f'(x',\pi)))^2 \tag{9}$$

$$\ell_{\text{goal}}(\bar{f}_g) := \frac{1}{|\bar{D}_{\text{goal}}|} \sum_{(x,a,r,x') \in \bar{D}_{\text{goal}}} (g(x) - 1)^2 \tag{10}$$

where we use $\phi(\bar{f}_g)(x, a) = \max(g(x), f(x, a))$ and for $x \notin \mathcal{X}^+$, $\bar{f}_g(x, a) = f(x,a)\mathbb{1}(a \neq a^+) + g(x)\mathbb{1}(a = a^+)$.

Using this loss, we define the two-player game of PSPI for the augmented MDP:

$$\max_{\pi \in \Pi} \min_{\bar{f}_g \in \bar{\mathcal{F}}} \bar{f}_g(d_0, \bar{\pi}) \tag{11}$$

$$\text{s.t.} \quad \ell_{\text{dyn}}(\bar{f}_g, \bar{f}_g; \bar{\pi}) - \min_{\bar{f}'_{g'} \in \bar{\mathcal{F}}} \ell_{\text{dyn}}(\bar{f}'_{g'}, \bar{f}_g; \bar{\pi}) \leq \epsilon_{\text{dyn}}$$

$$\ell_{\text{goal}}(\bar{f}_g) \leq 0$$

Notice $\bar{f}_g(d_0, \bar{\pi}) = f(d_0, \pi)$. Therefore, this problem can be solved using samples from $D$ without knowing $G$.

## B.4 ANALYSIS

**Covering number** We first define the covering number on the function classes $\mathcal{F}$, $\mathcal{G}$, and $\Pi$[13]. For $\mathcal{F}$ and $\mathcal{G}$, we use the $L_\infty$ metric. We use $\mathcal{N}_\infty(\mathcal{F}, \epsilon)$ and $\mathcal{N}_\infty(\mathcal{G}, \epsilon)$ to denote the their $\epsilon$-covering numbers. For $\Pi$, we use the $L_\infty$-$L_1$ metric, i.e., $\|\pi_1 - \pi_2\|_{\infty,1} := \sup_{x \in \mathcal{X}} \|\pi_1(\cdot|s) - \pi_2(\cdot|s)\|_1$. We use $\mathcal{N}_{\infty,1}(\Pi, \epsilon)$ to denote its $\epsilon$-covering number.

**High-probability Events** First, we show $\bar{Q}^{\bar{\pi}}$ has small empirical errors.

**Lemma B.7.** *With probability at least $1 - \delta$, it holds for all $\pi \in \Pi$,*

$$\ell_{dyn}(\bar{Q}^{\bar{\pi}}, \bar{Q}^{\bar{\pi}}; \bar{\pi}) - \min_{\bar{f}'_{g'} \in \bar{\mathcal{F}}} \ell_{dyn}(\bar{f}'_{g'}, \bar{Q}^{\bar{\pi}}; \bar{\pi}) \leq \epsilon_{dyn}$$

$$\ell_{goal}(\bar{Q}^{\bar{\pi}}) \leq 0$$

*where*[14]

$$\epsilon_{dyn} = O\left(\frac{\log\left(\mathcal{N}_\infty\left(\mathcal{F}, \frac{1}{|D_{dyn}|}\right)\mathcal{N}_\infty\left(\mathcal{G}, \frac{1}{|D_{dyn}|}\right)\mathcal{N}_{\infty,1}\left(\Pi, \frac{1}{|D_{dyn}|}\right)/\delta\right)}{|D_{dyn}|}\right)$$

---

[13]For finite function classes, the resulting performance guarantee will depend on $|\mathcal{F}|$, $|\mathcal{G}|$ and $|\Pi|$ instead of the covering numbers as stated in Theorem 3.5.

[14]Technically, we can remove $\mathcal{N}_\infty\left(\mathcal{G}, \frac{1}{|D_{\text{dyn}}|}\right)$ in the upper bound, but we include it here for a cleaner presentation.

*Proof.* Note $\bar{Q}^{\bar{\pi}} = \bar{f}_g$ for some $f \in \mathcal{F}$ and $g \in \mathcal{G}$ (Proposition B.6) and

$$0 = \mathbb{E}_{x,a \sim \mu_{\mathrm{dyn}}}[(\bar{Q}^{\bar{\pi}}(x,a) - \bar{\mathcal{T}}^{\bar{\pi}}\bar{Q}^{\bar{\pi}}(x,a))^2] = \mathbb{E}_{x,a \sim \mu_{\mathrm{dyn}}}[(\bar{Q}^{\bar{\pi}}(x,a) - 0 - \gamma\mathbb{E}_{x' \sim \bar{P}(\cdot|x,a)}[\phi(\bar{Q}^{\bar{\pi}})(x',\pi)])^2]$$

Following a similar proof of Theorem 8 of (Cheng et al., 2022), we can derive $\epsilon_{\mathrm{dyn}}$. On the other hand, $\ell_{\mathrm{goal}}(\bar{f}_g) = 0$ because the reward $R(x)$ is deterministic. $\qquad\square$

Nest, we show that with high probability the empirical error can upper bound the population error.

**Lemma B.8.** *For all $f \in \mathcal{F}, g \in \mathcal{G}$ satisfying*

$$\ell_{dyn}(\bar{f}_g, \bar{f}_g; \bar{\pi}) - \min_{\bar{f}'_{g'} \in \bar{\mathcal{F}}} \ell_{dyn}(\bar{f}'_{g'}, \bar{f}_g; \bar{\pi}) \leq \epsilon_{dyn}$$

$$\ell_{goal}(\bar{f}_g) \leq 0$$

*With probability at least $1 - \delta$, for any $f \in \mathcal{F}, g \in \mathcal{G}$*

$$\left\|\bar{f}_g(x,a) - \gamma\mathbb{E}_{x' \sim \bar{P}(\cdot|x,a)}[\max(g(x'), f(x', \pi))]\right\|_{\mu_{dyn}} \leq O\left(\sqrt{\epsilon_{dyn}}\right)$$

$$\|g(x) - 1\|_{\mu_{goal}} \leq O\left(\sqrt{\frac{\log\frac{\mathcal{N}_\infty\left(\mathcal{G}, \frac{1}{|D_{goal}|}\right)}{\delta}}{|D_{goal}|}}\right) =: \sqrt{\epsilon_{goal}}$$

*Proof.* This follows from Theorem 9 of (Cheng et al., 2022). $\qquad\square$

**Pessimistic Estimate**    We show the empirical value estimate found in equation 11 is pessimistic.

**Lemma B.9.** *Given $\pi$, let $\bar{f}_g^\pi$ denote the minimizer in equation 11. With high probability, $\bar{f}_g^\pi(d_0, \bar{\pi}) \leq Q^\pi(d_0, \pi)$*

*Proof.* By Lemma B.7, we have $\bar{f}_g^\pi(d_0, \bar{\pi}) \leq \bar{Q}_R^\pi(d_0, \bar{\pi}) = Q^\pi(d_0, \pi)$. $\qquad\square$

Next we bound the amount of underestimation.

**Lemma B.10.** *Suppose $x_0 \sim d_0$ is not in $G$ almost surely. For any $\pi \in \Pi$,*

$$Q^\pi(d_0, \pi) - \bar{f}_g^\pi(d_0, \bar{\pi})$$
$$\leq \mathbb{E}_\pi\left[\sum_{t=0}^{T-1} \gamma^t\left(\gamma\max(g^\pi(x_{t+1}), f^\pi(x_{t+1}, \pi)) - f^\pi(x_t, a_t)\right) + \gamma^T(R(x_T) - g^\pi(x_T))\right]$$

*Note that in a trajectory $x_T \in G$ whereas $x_t \notin G$ for $t < T$ by definition of $T$.*

*Proof.* Let $\bar{f}_g^\pi = (f^\pi, g^\pi)$ be the empirical minimizer. By performance difference lemma, we can write

$$(1 - \gamma)Q^\pi(d_0, \pi) - (1 - \gamma)\bar{f}_g^\pi(d_0, \bar{\pi})$$
$$= (1 - \gamma)\bar{Q}^\pi(d_0, \bar{\pi}) - (1 - \gamma)\bar{f}_g^\pi(d_0, \bar{\pi})$$
$$= \mathbb{E}_{\bar{d}^{\bar{\pi}}}[\bar{R}(x,a) + \gamma\bar{f}_g^\pi(x', \bar{\pi}) - \bar{f}_g^\pi(x,a)]$$

where with abuse of notation we define $\bar{d}^{\bar{\pi}}(x,a,x') := \bar{d}^{\bar{\pi}}(x,a)\bar{P}(x'|x,a)$, where $\bar{d}^{\bar{\pi}}(x,a)$ is the average state-action distribution of $\bar{\pi}$ in the augmented MDP.

In the above expectation, for $x \in G$, we have $a = a^+$ and $x^+ = (s^+, c)$ after taking $a^+$ at $x = (s, c)$, which leads to

$$\bar{R}(x,a) + \gamma\bar{f}_g^\pi(x', \bar{\pi}) - \bar{f}_g^\pi(x,a) = \bar{R}(x, a^+) + \gamma\bar{f}_g^\pi(x^+, \bar{\pi}) - \bar{f}_g^\pi(x, a^+) = R(x) - g^\pi(x)$$

For $x \notin G$ and $x \notin \mathcal{X}^+$, we have $a \neq a^+$ and $x' \notin \mathcal{X}^+$; therefore

$$\bar{R}(x,a) + \gamma\bar{f}_g^\pi(x', \bar{\pi}) - \bar{f}_g^\pi(x,a) = R(x) + \gamma\bar{f}_g^\pi(x', \bar{\pi}) - f^\pi(x,a)$$
$$\leq \gamma\max(g^\pi(x'), f^\pi(x', \pi)) - f^\pi(x,a)$$

where the last step is because of the definition of $\bar{f}_g^\pi$. For $x \in \mathcal{X}^+$, we have $x \in \mathcal{X}^+$ and the reward is zero, so

$$\bar{R}(x,a) + \gamma \bar{f}_g^\pi(x', \bar{\pi}) - \bar{f}_g^\pi(x,a) = 0$$

Therefore, we can derive

$$
\begin{aligned}
&(1-\gamma)Q^\pi(x_0, \pi) - (1-\gamma)\bar{f}_g^\pi(x_0, \bar{\pi}) \\
&\leq \mathbb{E}_{\bar{d}^{\bar{\pi}}}[\gamma \max(g^\pi(x'), f^\pi(x', \bar{\pi})) - f^\pi(x,a) | x \notin G, x \notin \mathcal{X}^+] + \mathbb{E}_{\bar{d}^{\bar{\pi}}}[R(x) - g^\pi(x) | x \in G]
\end{aligned}
$$

Finally, using Lemma B.2 we can have the final upper bound.

$\square$

## B.5 Main Result: Performance Bound

Let $\pi^\dagger$ be the learned policy and let $\bar{f}_g^{\pi^\dagger}$ be the learned function approximators. For any comparator policy $\pi$, let $\bar{f}_g^\pi = (f^\pi, g^\pi)$ be the estimator of $\pi$ on the data. We have.

$$
\begin{aligned}
&V^\pi(d_0) - V^{\pi^\dagger}(d_0) \\
&= Q^\pi(d_0, \pi) - Q^{\pi^\dagger}(d_0, \pi^\dagger) \\
&= Q^\pi(d_0, \pi) - \bar{f}_g^{\pi^\dagger}(d_0, \bar{\pi}^\dagger) + \bar{f}_g^{\pi^\dagger}(d_0, \bar{\pi}^\dagger) - Q^{\pi^\dagger}(d_0, \pi^\dagger) \\
&\leq Q^\pi(d_0, \pi) - \bar{f}_g^{\pi^\dagger}(d_0, \bar{\pi}^\dagger) \\
&\leq Q^\pi(d_0, \pi) - \bar{f}_g^\pi(d_0, \bar{\pi}) \\
&\leq \mathbb{E}_{\pi, P}\left[\sum_{t=0}^{T-1} \gamma^t (\gamma \max(g^\pi(x_{t+1}), f^\pi(x_{t+1}, \pi)) - f^\pi(x_t, a_t)) + \gamma^T(R(x_T) - g^\pi(x_T))\right] \\
&\leq \mathbb{E}_{\pi, P}\left[\sum_{t=0}^{T-1} \gamma^t |\gamma \max(g^\pi(x_{t+1}), f^\pi(x_{t+1}, \pi)) - f^\pi(x_t, a_t)| + \gamma^T|R(x_T) - g^\pi(x_T)|\right] \\
&\leq \mathfrak{C}_{\text{dyn}}(\pi)\mathbb{E}_{\mu_{\text{dyn}}}[|\gamma \max(g^\pi(x'), f^\pi(x', \pi)) - f^\pi(x,a)|] + \mathfrak{C}_{\text{goal}}(\pi)\mathbb{E}_{\mu_{\text{goal}}}[|g(x) - 1|] \\
&\leq \mathfrak{C}_{\text{dyn}}(\pi)\sqrt{\epsilon_{\text{dyn}}} + +\mathfrak{C}_{\text{goal}}(\pi)\sqrt{\epsilon_{\text{goal}}}
\end{aligned}
$$

where $\mathfrak{C}_{\text{dyn}}(\pi)$ and $\mathfrak{C}_{\text{goal}}(\pi)$ are the concentrability coefficients defined in Definition 3.4.

**Theorem B.11.** *Let $\pi^\dagger$ denote the learned policy of SDS + PSPI with datasets $D_{dyn}$ and $D_{goal}$, using value function classes $\mathcal{F} = \{\mathcal{X} \times \mathcal{A} \to [0,1]\}$ and $\mathcal{G} = \{\mathcal{X} \to [0,1]\}$. Under realizability and completeness assumptions as stated in Assumption 3.2 and Assumption 3.3 respectively, with probability $1 - \delta$, it holds, for any $\pi \in \Pi$,*

$$J(\pi) - J(\pi^\dagger) \leq \mathfrak{C}_{dyn}(\pi)\sqrt{\epsilon_{dyn}} + \mathfrak{C}_{goal}(\pi)\sqrt{\epsilon_{goal}}$$

*where*

$$\epsilon_{dyn} = O\left(\frac{\log\left(\mathcal{N}_\infty\left(\mathcal{F}, \frac{1}{|D_{dyn}|}\right)\mathcal{N}_\infty\left(\mathcal{G}, \frac{1}{|D_{dyn}|}\right)\mathcal{N}_{\infty,1}\left(\Pi, \frac{1}{|D_{dyn}|}\right)/\delta\right)}{|D_{dyn}|}\right),$$

*and,*

$$\epsilon_{goal} = O\left(\frac{\log\left(\mathcal{N}_\infty\left(\mathcal{G}, \frac{1}{|D_{goal}|}\right)/\delta\right)}{|D_{goal}|}\right)$$

*are statistical errors, and $\mathfrak{C}_{dyn}(\pi)$ and $\mathfrak{C}_{goal}(\pi)$ are concentrability coefficients which decrease as the data coverage increases.*

## C   Experimental details

### C.1   Hyperparameters and experimental settings

**IQL.**   For IQL, we keep the hyperparameter of $\gamma = 0.99$, $\tau = 0.9$, $\beta = 10.0$, and $\alpha = 0.005$ in Kostrikov et al. (2021), and tune other hyperparameters on the antmaze-medium-play-v2 environment and choose batch size = 1024 from candidate choices $\{256, 512, 1024, 2046\}$, learning rate = $10^{-4}$ from candidate choices $\{5 \cdot 10^{-5}, 10^{-4}, 3 \cdot 10^{-4}\}$ and 3 layer MLP with RuLU activating and 256 hidden units for all networks. We use the same set of IQL hyperparameters for both our methods and all the baseline methods included in Section 4.2, and apply it to all environments.

**RP.**   For naive reward prediction, we use the full context-goal dataset as positive data, and train a reward model with 3-layer MLP and ReLU activations, learning rate = $10^{-4}$, batch size = 1024, and training for 100 epochs for convergence. To label the transition dataset, we need to find some appropriate threshold to label states predicted as goals given contexts. We choose the percentile as 5% in the reward distribution evaluated by the context-goal set as the threshold to label goals in the antmaze-medium-play-v2 environment, from candidate choices $\{0\%, 5\%, 10\%\}$. Then we apply it to all environments. Another trick we apply for the reward prediction is that instead of predicting 0 for the context-goal dataset, we let it predict 1 but shift the reward prediction by -1 during reward evaluation, which prevents the model from learning all 0 weights. Similar tricks are also used in other reward learning baselines.

**UDS+RP.**   We use the same structure and training procedure for the reward model as RP, except that we also randomly sample a minibatch of "negative" contextual transitions with the same batch size for a balanced distribution, which is constructed by randomly sampling combinations of a state in the trajectory-only dataset and a context from the context-goal dataset. To create a balanced distribution of positive and negative samples, we sample from each dataset with equal probability. For the threshold, we choose the percentile as 5% in the reward distribution evaluated by the context-goal set as the threshold to label goals in the antmaze-medium-play-v2 environment, from candidate choices $\{0\%, 5\%, 10\%\}$. Then we apply it to all environments.

**PDS.**   We use the same structure and training procedure for the reward model as RP, except that we train an ensemble of 10 networks as in Hu et al. (2023). To select the threshold percentile and the pessimistic weight $k$, we choose the percentile as 15% in the reward distribution evaluated by the context-goal set as the threshold to label goals from candidate choices $\{0\%, 5\%, 10\%, 15\%, 20\%\}$, and $k = 15$ from the candidate choices $\{5,10,15,20\}$ in the antmaze-medium-play-v2 environment. Then we apply them to all environments.

**SDS (ours).**   We do not require extra parameters other than the possibility of sampling from the real and fake transitions. Intuitively, we should sample from both datasets with the same probability to create an overall balanced distribution. Empirically, we also find that the balance distribution generates the best result.

### C.2   Reward model evaluation

For reward learning baselines, we evaluate the learned reward model: we construct the positive dataset from context-goal examples, and the negative dataset from the combination of the context set and all states in the trajectory-only data, using the oracle context-goal function defined in the environment to filter out positive ones. We then evaluate the predicted reward on both positive and negative datasets, generating boxplots to visualize the distributions of the predicted reward for both datasets. The purpose of the reward model evaluation is to showcase whether the learned reward function can successfully capture context-goal relationships.

## D   More reward model evaluations

Here we present boxplots for reward models with experimental setups in Section 4.3, 4.4 and 4.5.

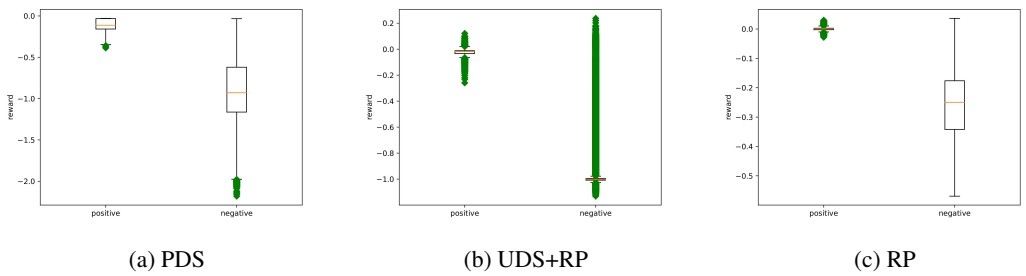

(a) PDS            (b) UDS+RP            (c) RP

Figure 3: Reward model evaluation for the medium-diverse environment in Section 4.3. Green dots are outliers.

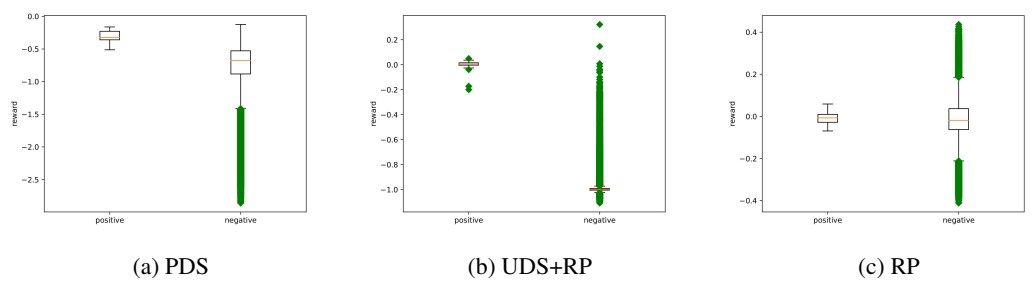

(a) PDS            (b) UDS+RP            (c) RP

Figure 4: Reward model evaluation for the umaze-diverse environment in Section 4.3. Green dots are outliers.

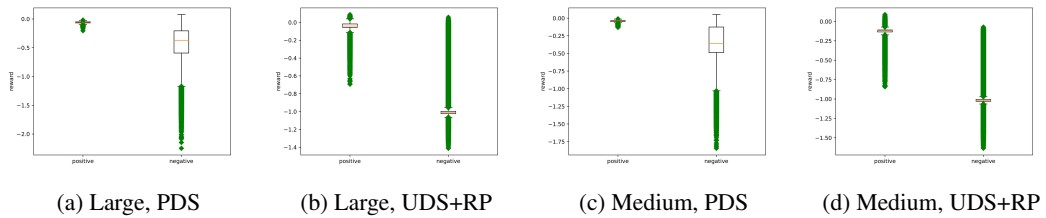

(a) Large, PDS     (b) Large, UDS+RP     (c) Medium, PDS     (d) Medium, UDS+RP

Figure 5: Reward model evaluation for Four Rooms in Section 4.4. Green dots are outliers.

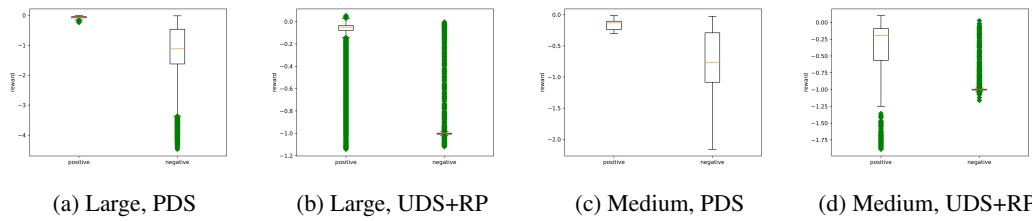

(a) Large, PDS     (b) Large, UDS+RP     (c) Medium, PDS     (d) Medium, UDS+RP

Figure 6: Reward evaluation for Random Cells in Section 4.5 (the test context distribution is the same as training). Green dots are outliers.

