# OpenReview forum: "Simple Data Sharing for Multi-Tasked Goal-Oriented Problems"
_ICLR.cc/2024/Conference — Submitted to ICLR 2024_

### Official Review · Reviewer_Mndx · 2023-10-30

**Soundness:** 2 fair
**Presentation:** 3 good
**Contribution:** 2 fair
**Rating:** 5
**Confidence:** 3

**Summary:**

This paper tackles Contextual Goal-Oriented  (CGO) problems, which generalizes the goal-oriented (GO) setting and considers a context variable (e.g. natural language instructions) which specifies a particular goal in a multi-task RL setting. They propose Simple Data Sharing (SDS) which can provably solve CGO problems by utilizing (1) an unsupervised dataset of transitions and (2) a dataset which additionally includes context-goal pairs. SDS first defines an action-augmented MDP which resolves the inconsistent dynamics in the two datasets and where the optimal policies are also optimal in the original CGO. Then, they propose a data augmentation technique for integrating the two datasets in a principled manner, where this process can be used as a preprocessing step before running offline RL algorithms. Their approach is evaluated on AntMaze.

**Strengths:**

1. The offline CGO problem is important and timely given the current trend of integrating natural language, foundation models and RL, where large-scale robotics datasets have already started to be released (e.g. RT-X).
2. The authors propose a simple and theoretically sound technique for augmenting the unsupervised dataset of transitions and supervised datasets with context-goal pairs. This setting is well motivated in real-world applications where the unsupervised datasets may be easier to obtain.
3. The connection between the original offline CGO problem and the action-augmented MDP is interesting.

**Weaknesses:**

1. As the authors themselves outline in the conclusion, only AntMaze was considered. While I appreciate that the authors were open about the limited experiments, I think comprehensive experiments are something necessary to establish the strength of SDS rather than it being a “interesting future direction” As a comparison, MAHALO considers more settings on MuJoCo (D4RL datasets) as well as MetaWorld.
2. While it is true that unsupervised datasets such as task-agnostic play data can be easier to obtain than supervised datasets, SDS requires the dynamics of the dataset to cover goal-reaching trajectories. This is essentially suggesting that SDS always requires optimal trajectories, since we can just add a goal-reaching indicator at the last step of that trajectory even if it is unsupervised initially. This assumption may not scale to more complex GCRL problems if the goals are harder to reach and only near-optimal agents can achieve that goal.
3. The introduction motivates SDS by claiming that Offline GCRL algorithms such as GoFar (Ma et, al. 2022) “can fail when the predicted goal is not reachable from the initial state.” However, this claim is also not evaluated in the experiments. Overall, there are a lack of baselines which were included in the related work section but not compared in the experiments.
4. Similarly, MAHALO was not considered as a baseline for the experiments although it was compared to SDS in the theoretical results.
5. While the theoretical results in Theorem 3.1 are nice,  it is not immediately clear what the novel contributions are. While I am not too familiar with the literature on statistical RL and I may be missing something, I think the authors should be explicit about what novel techniques are used to extend theorems from previous work, especially if the theoretical results are the main contribution of the paper.
6. (Minor) The indices for theorems in the Appendix and main text are not aligned and there are no references e.g. Theorem B.11 and Theorem 3.1.

**Questions:**

1. What Is the main difficulty in obtaining negative data (context-non-goal pairs) which is described as a limitation of MAHALO? What makes the unsupervised dynamics data easier to obtain?
2. Please clarify the points raised in the Weaknesses.

---

> ### Author Response · Authors · 2023-11-21
> **Rebuttal**
>
> **1. "As the authors themselves outline in the conclusion, only AntMaze was considered. While I appreciate that the authors were open about the limited experiments, I think comprehensive experiments are something necessary to establish the strength of SDS rather than it being a “interesting future direction” As a comparison, MAHALO considers more settings on MuJoCo (D4RL datasets) as well as MetaWorld."**
>
> We chose AntMaze because it is a popular benchmark and it is easy to adapt its environments to construct offline CGO problems. (On the other hand, locomotion mujoco problems in D4RL are not goal oriented). . Besides the 3 variations, we also add goal prediction as an extra baseline and experiment with new perturbation settings (Section 4.4 and 4.5) when constructing the context-goal dataset, which helps to show the efficacy of the method in a better way.
>
> **2. "While it is true that unsupervised datasets such as task-agnostic play data can be easier to obtain than supervised datasets, SDS requires the dynamics of the dataset to cover goal-reaching trajectories. This is essentially suggesting that SDS always requires optimal trajectories, since we can just add a goal-reaching indicator at the last step of that trajectory even if it is unsupervised initially. This assumption may not scale to more complex GCRL problems if the goals are harder to reach and only near-optimal agents can achieve that goal."**
>
> We respectfully disagree with the claim that “This is essentially suggesting that SDS always requires optimal trajectories, since we can just add a goal-reaching indicator at the last step of that trajectory even if it is unsupervised initially” is not true. We’re sorry that our original manuscript might have created this misconception. We would like to clarify that our method does not directly add a goal reaching indicator at the last step **since the goals context-goal only dataset might not appear in the dynamics dataset, which is the case in our newly added experiments in Section 4.4 and 4.5**.
>
> Theoretically, we only require 1) the distribution generating the dynamics dataset "covers" a feasible path to the target goal set via the generalization ability of the learned value function (which includes stitching) and 2) the context-goal distribution "covers" some feasible goal via the generalization ability of function approximation (which does not require any access to non-goal examples), which is scalable to complex settings. We also add additional explanations in our assumptions and theorems in the revision.
>
> **3. "While the theoretical results in Theorem 3.1 are nice, it is not immediately clear what the novel contributions are. "**
>
>
> The main technical contribution here is to show that SDS which is based on a fictitious action can be implemented with PSPI by using function approximators and their assumptions based on the original MDP without the fictitious action. The details are given in Appendix B.3 and Appendix B.4 ( Lemma B.10): We show that by carefully analyzing the Bellman equation of the action-augmented MDP, we can translate the needed assumptions back those defined in the original MDP, so we have Assumption 3.2-3.4 defined in natural terms wrt the original MDP. These techniques are novel and critically leverage the structure of CGO and SDS. The statistical analyses in Appendix B.4 (High-probability Events) are standard
>
> Having the assumptions made in terms of the original MDP (not the augmented MDP) is important to establishing a formal theoretical result. Otherwise, the assumptions may be vacuous as they cannot be compared with others who do not introduce the augmented MDP.  We will highlight the importance of these contributions better in the final draft.
>
>
> **4. "The introduction motivates SDS by claiming that Offline GCRL algorithms such as GoFar (Ma et, al. 2022) “can fail when the predicted goal is not reachable from the initial state.” However, this claim is also not evaluated in the experiments. Overall, there is a lack of baselines which were included in the related work section but not compared in the experiments."**
>
> Thanks for your suggestion! **We also add the goal prediction baseline in Section 4.4 and 4.5.**
>
> **5. "Similarly, MAHALO was not considered as a baseline for the experiments although it was compared to SDS in the theoretical results."**
>
> We did initial trials using MAHALO in the AntMaze environments, and we found that the performance is not as good as PDS+IQL or goal prediction + IQL in larger mazes. One reason may be that the public MAHALO implementation is based on ATAC and PSPI, which do not have published results and hyperparameters for  Antmaze. Also, MAHALO requires adversarial training to learn a pessimistic reward function which might pose extra difficulty in the training.

---

> > ### Author Response · Authors · 2023-11-21
> > **Rebuttal (continued)**
> >
> > **6. "What Is the main difficulty in obtaining negative data (context-non-goal pairs) which is described as a limitation of MAHALO? What makes the unsupervised dynamics data easier to obtain?"**
> >
> > We discuss why unsupervised datasets and context-goal datasets are commonly available in the introduction: In robotics, task-agnostic play data (unsupervised data) can be obtained at scale since it does not require any reward labels. Positive data (context-goal pairs) would be easy to obtain as "successful samples" as in some instruction datasets, while negative data under the CGO setting would be "failure samples" which are generally harder to obtain in common instruction datasets.

---

### Official Review · Reviewer_wCFi · 2023-10-31

**Soundness:** 2 fair
**Presentation:** 2 fair
**Contribution:** 2 fair
**Rating:** 5
**Confidence:** 2

**Summary:**

This work introduces the concept of Contextual Goal Oriented (CGO) problems, which are a specific type of multi-task decision problems that involve reaching goals while following instructions. The authors propose a framework for designing agents that can solve such tasks by leveraging unsupervised interaction data and a supervised dataset of context-goal pairs. They demonstrate that their approach outperforms traditional offline reinforcement learning methods in simulated CGO problems, providing near-optimal context-conditioned policies.

**Strengths:**

The application of a relatively simple technique for contextual multi-task offline RL, alongside existing offline RL methodologies, is highly valuable. Additionally, the theoretical analysis of performance in this context is solid.

**Weaknesses:**

The applied assumptions in the paper are quite strong, and the experimental validation of the proposed methodology is limited. Consequently, it is challenging to determine whether the proposed approach would work in more complex and practical scenarios.

**Questions:**

Is it possible to generate policies for achieving any arbitrary goal state, and if so, what are the principles for generalizing across different goals? Given a limited set of goal contexts, what needs to be learned to derive action plans for arbitrary goals? If so, does this study propose a specific model structure or learning technique for learning such information?

It seems that the scope of the problem addressed in this research is quite limited. Can the proposed methodology be applied to other problem classes that are more general in nature?"

---

> ### Author Response · Authors · 2023-11-21
> **Rebuttal**
>
> ### Weaknesses
> **1. “The applied assumptions in the paper are quite strong”**
>
> As opposed to some misunderstandings of our assumptions raised by the reviewers, we highlight that we only require 1) the distribution generating the dynamics dataset "covers" a feasible path to the target goal set and 2) the context-goal distribution "covers" some feasible goals. We remark that "covering" here does not mean the such a path or a goal has to appear in the datasets or the data have to be generated only by goal reaching policies (e.g., it suffices that data can be stitched together to form a goal reaching path). Instead, it is a distribution-level coverage requirement; especially we will present coverage conditions based on the generalization ability of function approximators used in learning, which is weaker than requiring coverage conditions based on non-zero density.
>
> **2. "the experimental validation of the proposed methodology is limited"**
>
> We chose AntMaze because it is a popular benchmark and it is easy to adapt its environments to construct offline CGO problems. (On the other hand, locomotion mujoco problems in D4RL are not goal oriented). Besides the 3 variations, we also add goal prediction as an extra baseline and experiment with new perturbation settings (Section 4.4 and 4.5) when constructing the context-goal dataset, which helps to show the efficacy of the method in a better way.
>
> ### Questions
>
> **1. "Is it possible to generate policies for achieving any arbitrary goal state, and if so, what are the principles for generalizing across different goals?"**
> **2. "Given a limited set of goal contexts, what needs to be learned to derive action plans for arbitrary goals? If so, does this study propose a specific model structure or learning technique for learning such information"**
>
> 1&2 are highly related so we answer them in one piece. In our CGO setting, the MDP is context-dependent, which includes goal-dependency as a special case (i.e. the traditional goal oriented problem is the case when  context=goal). As opposed to learning a context dependent reward function/goal predictor in the baseline methods, we directly learn context-dependent value functions and policies via adding the fictitious transitions and fictitious actions as stated in the paper. We can generate policy for arbitrary contexts since it takes context as input, making it natural to generalize across different contexts.

---

> > ### Comment · Reviewer_wCFi · 2023-11-23
> > **Thank you for your response.**
> >
> > Thank you for your response. I appreciate the additional explanation regarding the assumption.
> >
> > I don't believe that simply using contextual information about the goal as input will improve the generalization performance for the goal. I was curious if there is an architecture or learning approach that handles information about the goal and, through this, enhances the generalization ability for arbitrary goals.
> >
> > My opinion remains unchanged and it would be nice to see this paper validated in more diverse environments and complex tasks.

---

> > > ### Author Response · Authors · 2023-11-23
> > >
> > > Thanks for your response. Just to clarify, we do not “simply use contextual information about the goal as input” which is another misunderstanding of our work. We do have an "architecture or learning approach that handles information about the goal": the context-goal relationship is learned using the fictitious action and transition (from the goal states to the fictitious terminal state given the context) and we are directly learning the value function and policy based on the context, as described in the paper in the method section. We could this generalize to unseen context and also infer feasible under that context goals due to the generalization ability of our learned context dependent value function and policy.

---

### Official Review · Reviewer_Fsyr · 2023-11-04

**Soundness:** 2 fair
**Presentation:** 2 fair
**Contribution:** 2 fair
**Rating:** 3
**Confidence:** 2

**Summary:**

This paper considers a setting of contextual goal oriented problems, where the agent has access to an unsupervised transition dataset and goal context-state pairs. The authors develop a simple data sharing (SDS) method after constructing an augmented MDP with the auxiliary state and action. Theoretical analysis of a variant employing SDS is provided to show its optimality. Experimental results on established AntMaze datasets also show the effectiveness of SDS+IQL compared to designed baselines.

**Strengths:**

1. This paper introduces the contextual goal oriented (CGO) problem, which may shed some light on practical problems in robotics or navigation, as illustrated in the introduction, where the task-agnostic data and context-goal mapping are collected separately.
2. The proposed approach of simple data sharing tackles this problem in a pre-processing way that can be integrated with off-the-shelf offline RL methods.
3. The authors analyze the theoretical effectiveness of SDS+PSPI and provide a strong performance of SDS+IQL mainly in the AntMaze environment with different levels of context distributions.

**Weaknesses:**

1. Although some discussions are involved in the introduction, I think this paper still lacks explanations about the CGO setting. I doubt the coexistence of separate unsupervised data and context-goal data might not be so difficult to combine. For example, in the navigation setting mentioned by the authors, it is easy to derive a goal position from the actual state and vice versa. With a predefined context distribution, we can label a state accomplishing a given context as the goal state with ease.
2. In Section 3.1, the authors see the key problem that "some $s\in G_c$ in the $D_\text{goal}$ dataset is also observed in the dynamics dataset." However, I speculate that the process of Algorithm 1 cannot tackle this issue as the transition of a goal state $x$ to $x'$ where $x' \neq x^+$ will still remain. The problem can only be alleviated with the mentioned apporach of "equally balancing the samples $\bar{D}_\text{dyn}$ and $\bar{D}_\text{goal}$" (in section 3.2).
3. A confusing point is that the authors provide theoretical analysis on SDS+PSPI but do not use this variant for experiments (they use SDS+IQL instead).

Some minor issues:

1. In Line 7 of Algorithm 1, should the created transition here be incorperated with $a$ rather than $a^+$? Otherwise I do not observe any usage of $a$ from unsupervised data.
2. In the "data assumption" part of Section 2, the definitions between $\mu_\text{dyn}(s, a, s')$ and $\mu_\text{dyn}(s' \mid s, a)$, and between $\mu_\text{goal}(s, c)$ and $\mu_\text{goal}(s \mid c)$ seem to be abused.

**Questions:**

1. The authors should include introductions of related data sharing baselines in the preliminaries or related work section, e.g., CDS, PDS and UDS partially used as baselines.
2. What is the applicable domain of CGO problems? As mentioned in the weaknesses, why can't we relabel the unsupervised data for a targeted task as the context can be usually derived from the state, since the context $c$ in this setting remains unchanged? I may foresee an issue that this derivation may not be easy for language instructions. However, texts as contexts are not involved in the experiments. The authors also mention that they can use "an oracle function to tell whether a state is within the goal set" (in Section 4.1).
3. Why do the experiments replace 0/1 rewards with -1/0 rewards? Are they intrinsically different or empirically different?
4. Why not use the theoretically sound SDS+PSPI method for experiments?
5. I'm confusing about the reward prediction approach adopted to the baselines. How can we train a reward model with only using postive data (from $D_\text{goal}$)? For the usage of the reward model, can you ellaborate the meaning of "choosing the percentile as 5% in the reward distribution evaluated by the context-goal set as the threshold to label goals" in Appendix C.1?
6. In Section 4.3, what is the reason for the observation that "for PDS, we can observe that the reward distribution for positive and negative samples are better separated in the large one than the medium one?"
7. What are main differences among the three experimental settings (from Section 4.3 to 4.5)? I observe a different range of context distribution but it seems that all test contexts are either fully in-distribution or partially in-distribution (Table 4) but not include real out-of-distribution cases.

---

> ### Author Response · Authors · 2023-11-21
> **Rebuttal**
>
> **1. "I doubt the coexistence of separate unsupervised data and context-goal data might not be so difficult to combine… we can label a state accomplishing a given context as the goal state with ease. " (Similar: why can't we relabel the unsupervised data for a targeted task as the context can be usually derived from the state…. The authors also mention that they can use "an oracle function to tell whether a state is within the goal set".)**
>
>
> While what the reviewer mentioned is applicable in some special cases, such a hindsight construction is not possible for offline CGO problems in general. Here are the reasons:  **1) the test context which might not even exist in the context-goal data so we cannot directly label goal states (e.g. when the context space is continuous); and 2) the exact same states in the context-goal dataset could never appear in the unsupervised dataset.** Therefore, an example of this kind is our third experimental setup in Section 4.5, where the test context is randomly sampled so it could never appear in the training set (but potentially still under coverage through function approximation). In the revision, we also add new experimental results in our draft where we perturb the context-goal dataset **(Section 4.4 and 4.5) such that the goals in the context-goal dataset never appear in the unsupervised dataset, and show that our method still works well**. Finally, as in our footnote for the oracle function, it is only used to generate context-goal dataset for the sake of problem construction; this information is not revealed to the learning algorithms and with further perturbation of the context-goal dataset it is also not possible to  “label a state accomplishing a given context as the goal state with ease”.
>
> **2. "In Section 3.1, the authors see the key problem that "some s \in G_c in the dataset is also observed in the dynamics dataset." However, I speculate that the process of Algorithm 1 cannot tackle this issue as the transition of a goal state x to x’ where will x’ \neq x^+ still remains. The problem can only be alleviated with the mentioned approach of "equally balancing the samples $\bar{D}\text{dyn} \bar{D}\text{goal}$" (in section 3.2)."**
>
> The claim that “Algorithm 1 cannot tackle this issue as the transition of a goal state x to x’ where will x’ \neq x^+ still remains” is a misunderstanding, **since a^+ is not in the original action space**. For example, x is a goal state, x’ is some non-goal state and there is some transition (x,a,x’) in the original dynamics dataset, and we add (x, a^+, x^+) as fictitious transitions. As a result, Q(x,a^+) would be high while Q(x,a) will be low, so V^*(x) will still be high. In general, we introduce an extra fictitious action a^+ for x to reach x^+, which is outside the original action space in the unsupervised dataset, so there is no such conflict in dynamic programming when we compute the optimal value function via argmax over all possible actions (including both real and fictitious actions). If such fictitious action and transition exists with non-zero probability, it will naturally be the highest Q value over all possible actions (e.g., via expectile regression for IQL,which does not necessarily require balancing the samples).
>
> **3. "A confusing point is that the authors provide theoretical analysis on SDS+PSPI but do not use this variant for experiments (they use SDS+IQL instead). Why not use the theoretically sound SDS+PSPI method for experiments?"**
>
> Thank you for raising this point. We choose IQL since it is a popular practical algorithm to solve AntMaze environments, making it easier to compare with the literature and other baselines. Including SDS+IQL in the experiments also shows that the idea of SDS is indeed general, not just limited to PSPI.
>
> **4. "Why do the experiments replace 0/1 rewards with -1/0 rewards? Are they intrinsically different or empirically different?"**
>
> To keep our method comparable to the context agnostic baseline, we follow the tradition to use -1/0 reward in AntMaze experiments as in the original IQL paper. The -1/0 rewards are just a scaled and shifted version of the 1/0 rewards (specifically, first scale the reward of 1/0 by $\frac{1}{1-\gamma}$ to the reward of $\frac{1}{1-\gamma}$ / 0 and then shift the reward at each state by -1; then we get the -1/0 reward). Therefore these two reward systems induce the same policy ranking, and our theory applies.

---

> ### Author Response · Authors · 2023-11-21
> **Rebuttal (continued)**
>
> **5. "How can we train a reward model with only using positive data (For the usage of the reward model, can you elaborate the meaning of "choosing the percentile as 5% in the reward distribution evaluated by the context-goal set as the threshold to label goals" in Appendix C.1?"**
>
> As mentioned in 4.2, we use PDS to train a pessimistic reward function with ensembles of 10 reward functions, which measures the uncertainty of the value function prediction, making it possible to learn a pessimistic reward as introduced in PDS. Since the uncertainty term is subtracted from the reward predicting, making it a continuous reward prediction and we need to convert it back to -1/0 to label the goals, we need to select some certain threshold. So we use the tradition in outlier detection where we choose some percentile of the predicted values from the training distribution as the threshold.
>
> Nonetheless, we also note that the pessimistic rewards learned in this way, and the more naive version RP, are not perfect and can be far away from the true rewards. Nonetheless, we saw in the experiments, IQL with these imperfect rewards can still have fairly good performance in some datasets. Such a robust behavior to reward imperfection has been recently observed and proved for offline RL in the literature [Li et al. 2023, Shin et al. 2023].
>
> Li, Anqi, et al. "Survival Instinct in Offline Reinforcement Learning." arXiv preprint arXiv:2306.03286 (2023).
>
> Shin, Daniel, Anca D. Dragan, and Daniel S. Brown. "Benchmarks and algorithms for offline preference-based reward learning." arXiv preprint arXiv:2301.01392 (2023).
>
>
> **6. "In Section 4.3, what is the reason for the observation that "for PDS, we can observe that the reward distribution for positive and negative samples are better separated in the large one than the medium one?"**
>
> For different environments, we have different context-goal dataset and trajectory only dataset. The performance of PDS relies on how well it can capture the pessimism correctly in our test context distribution given the training data. And we observe that although the medium maze is a smaller problem than the large maze, the training distribution is harder for PDS to separate the positive and negative examples for the test distribution.
>
> **7. "What are the main differences among the three experimental settings (from Section 4.3 to 4.5)? I observe a different range of context distribution but it seems that all test contexts are either fully in-distribution or partially in-distribution (Table 4) but not include real out-of-distribution cases."**
>
> The main differences are shown in Fig 1 and also discussed in 4.1: 1) the easy case where goal sets are very similar such that it could be approximately solved by a context agnostic policy, 2)We have finite contexts and the goal sets from different contexts are distinct and do not overlap  3)The contexts are continuous and randomly sampled, the goal sets from different contexts are different but could share some overlap, making a complex CGO problem. We do not aim to show out-of-distribution generalization for context distributions. However, the learned policy could enjoy some out-of-distribution generalization which would be an interesting future direction.

---

> ### Comment · Reviewer_Fsyr · 2023-11-22
>
> Thanks for the authors' rebuttal that addresses some of my questions. After reading the revision and other reviews, I still have concerns about the discrepancy between theories and practical solutions. Besides, I think the authors should consider changing the experiment design since current setup of different goal levels does not help show different properties of SDS and also lack diverse benchmarks. I will keep my score.

---

### Official Review · Reviewer_i3TD · 2023-11-05

**Soundness:** 1 poor
**Presentation:** 2 fair
**Contribution:** 3 good
**Rating:** 3
**Confidence:** 4

**Summary:**

This paper considers the goal-conditioned offline RL problem where an agent needs to learn to reach given goals given offline training data. They assume that the dataset contains environment trajectories (excluding rewards and termination flags), $D_{dyn}$, and goal labels for some states, $D_{goal}$. They then propose combining the given dataset to produce a trajectories dataset that includes rewards and termination flags, such that any offline algorithm can be used to learn policies. Precisely, their approach (SDS) adds an absorbing action and an absorbing state to the original trajectories, then gives rewards of 1 for the absorbing action in states with goal labels ($D_{goal}$) and rewards of 0 for the absorbing action in all states in $D_{dyn}$. They then provide a theorem bounding the optimality of policies learned from the modified dataset, and provide experiments comparing their approach with other offline RL methods.

**Strengths:**

- The paper is mostly well-written and investigates an important problem. The specific proposed approach for learning goal-conditioned policies given a trajectory dataset and only positive examples of goal-state labels is novel and interesting.
- The proposed algorithm is theoretically grounded by showing that the set of optimal value functions is not unchanged by the approach.
- The paper shows numerous empirical results comparing SDS + OfflineRL with a couple OfflineRL baselines in continuous control tasks. Impressively, SDS + OfflineRL is either comparable or outperforms prior OfflineRL works.

**Weaknesses:**

**MAJOR**

1- **Theory**:

  - The theoretical assumptions seem imprecise or incorrect. For example, the authors say that the state, action, and context spaces can be continuous. If they are continuous (say $|\mathcal{A}|=|\mathcal{S}|=|\mathbb{R}|$), then $|\Pi|$ is extremely large (e.g contains $|\mathcal{A}|^{|\mathcal{S}|}=\aleph_1^{\aleph_1}$ deterministic policies). In this case $\epsilon_{dyn}$ will be extremely large, making the RHS of the bound in Theorem 3.1 extremely large.

  - If I understand Theorem 3.1 correctly, then the right-hand side of the bound is almost always greater than $V_{max}=\frac{1}{1-\gamma}$ (since the rewards are only non-zero at goal states and are 1 otherwise). Since this would make the bound useless, I am probably misunderstanding the meaning of each term on the RHS. It would have been helpful if the authors explained step-by-step each term of the bound and how that leads to a small regret with high probability (maybe using an example domain)

  - The motivation for why Algorithm 1 is expected to work is not given, which makes it hard to get an intuition for its general applicability. In fact, the algorithm looks like it shouldn't work, since it gives zero rewards for all states in $D_{dyn}$, potentially including goal states that were given rewards of 1 in $D_{goal}$. It is not clear why this reward scheme makes sense.

  - The presentation of the theoretical results is very poor (mainly Section 3.3).

    - There is barely any explanation of what the Theorem, Assumptions, and Definitions are saying. For example, Definition 3.4 defines "generalized concentrability" out of the blue and does not explain what it is saying and how it is relevant. It is also not clear why it uses the same symbols that were defined for just "concentrability" in Theorem 3.1.

    - Some symbols or functions like $a^+$, $g$, $\mathcal{F}$, and $\mathcal{G}$ are defined but not explained. What are they supposed to represent? For example, I am assuming $a^+$ represents a terminating action similar in behavior to terminating sets in Options. For another example, Theorem 3.1 suggests $\mathcal{F}$ and $\mathcal{G}$ are sets of value functions, but Assumption 3.2 suggests $\mathcal{G}$ is a set of reward functions. It is also not clear what $R$ in this assumption is (I am just assuming it is a reward function). It is also unclear what MDP $Q^\pi$ and $R$ are defined for in this assumption.

    - There is a general sense of scattered explanations that makes the paper hard to read. For example, Section 3.3 jumps straight into Theorem 3.1 which is about SDS + PSPI without even explaining PSPI. Only after is PSPI explained in Section 3.3.1.


2- **Experiments**:

  - All the experiments are in a single domain (the D4RL AntMaze). While the 3 variations in goal distributions are useful, comparisons with baselines in other domains (e.g another D4RL domain) would have helped get a better sense of the general applicability of the proposed approach

  - The paper reports means and standard deviations over only 5 seeds. These are not enough to support the strong empirical claims of the paper.

3- **Related Works**:

  - There is no related works section in the main paper. I think moving the related works to the appendix is going a bit too far. The related works section is important to contextualise the contributions of this work in relevant literature properly. At least a brief related works section could have been included in the main paper (leaving the expanded version for the appendix).

3- **Limitations**:

  - The proposed approach is only applicable to goal-conditioned tasks with rewards of 1 only at goal states and zero rewards otherwise.

  - The paper makes very strong assumptions about the offline dataset. Mainly, they assume that the goals dataset ($D_{goal}$) contains all the goals the agent will encounter and that the trajectories dataset ($D_{dyn}) contains trajectories leading to those goals.


**MINOR**

- What is the meaning of "the goal sets can overlap but their intersection is empty'
- Figure 1.c is unclear. What are the overlapping goals?
- "Medium-play and large-play datasets" are never explained. A brief explanation would have helped with readability.
- Page 7 "different ..."

**Questions:**

It would be great if the authors could address the major weaknesses I outlined above. I am happy to increase my score if they are properly addressed, as I may have misunderstood pieces of paper.

Additionally,

- I suggest the authors follow the Theorem, Assumptions, and Definitions with clear explanations of what they are saying.
- It may help readability to start section 3.3 with 3.3.1 instead of the Theorem.

**### POST REBUTTAL ###**

Thank you to the authors for the time and effort they spent providing clarifications to my concerns. I have carefully read their response to my review and all the other reviewers. Their response has indeed addressed some of my concerns, precisely: Theory 1,2,4 and Limitations 2. The revised paper is also much clearer than before. Unfortunately, some of my major concerns about the theory, experiments, and related works remain.
- The authors completely ignored my concern about the related works section.
- My concern about the experiments remains. I think the authors should also demonstrate their approach in other domains.
- Theory 3 does not address my concern that the reward scheme used in Algorithm 1 is potentially problematic. If $(s_0,a_0,s_1),...,(s_t,a_t,s_{t+1}),...,(s_{T-1},a_{T-1},s_T)$ is a trajectory in $D_{dyn}$ and $(s_t,c)$ is in $D_{goal}$, then $\bar D_{dyn}$ will contain $((s_t,c), a^+, 0, (s_{t+1},c))$ but $\bar D_{goal}$ will contain $((s_t,c), a^+, 1, (s^+,c))$.
  - This contradiction in the rewards and transitions of $\bar D_{dyn}$ and $\bar D_{goal}$ make the resulting offline-RL dataset ill-defined.
  - $\bar D_{dyn}$ also doesn't satisfy the given augment MDP definition. I.e. $a^+$ must lead to $s^+$ with reward of 1, not  $s_{t+1}$ with reward of 0.
  - Finally, I assume a transition with the original action $a_t$ is also added to $\bar D_{dyn}$ (otherwise the generated offline dataset $\bar D_{dyn} \cup \bar D_{goal}$ will only contain the $a^+$ action for all transition samples). However, the handling of that action, e.g. $((s_t,c), a_t, ???, (s_{t+1},c))$, is not mentioned in Algorithm 1 nor anywhere else in the paper.
- Appendix B1 doesn't help either as it is riddled with imprecise and unjustified statements. For example,
  - $\bar V^{\bar \pi}(s) := \bar Q^{\bar \pi}(x,\bar \pi)$ is imprecise and inconsistent in notations. Maybe the authors instead meant  $\bar V^{\bar \pi}(x) := \bar Q^{\bar \pi}(x,\bar \pi(x))$, but I can't be certain since they then use this in their subsequent statements/proofs.
  - They authors state that Lemma B.2. is obvious and hence offer no proof. This is unjustified. Remark B.1. is also unjustified and potentially ill-defined (it is unclear what $x$ is an element of here, e.g ${X}, \bar {X}, {X}^+$, and also $\pi$ is not defined for all $a \in \bar A$).

In general, I think the approach proposed in this paper is promising and probably sound since the empirical results are also promising, but the authors just need to improve their experiments and revise their theoretical presentation thoroughly to make sure it is clear and indeed sound. Hence, I decreased my score from a 5 to a 3, and have increased my confidence from a 3 to a 4.

---

> ### Author Response · Authors · 2023-11-21
> **Rebuttal**
>
> ### Theory:
> **1.“The theoretical assumptions seem imprecise or incorrect …the authors say that the state, action, and context spaces can be continuous.”**
>
> Thank you first for raising this point. As we discussed in the footnote of the theorem, we only use the discrete notation (i.e. writing the cardinality of function and policy classes) for simplicity and “We state a more general result for non-finite function classes in the appendix”. In Appendix B4, **we introduce the definition of covering number and replace |\Pi| with the covering number of the function class**.
>
> **2. “It would have been helpful if the authors explained step-by-step each term of the bound and how that leads to a small regret with high probability”**
>
> Certainly, we can interpret the upper bound in the theorem as follows: the statistical errors would decrease as we have more data from $\mu_{goal}$ and $\mu_{dyn}$; then with the finite coefficients, the final upper bound would also decrease. Take $\pi = \pi^*$ as an example. For the coefficients to be finite,  it means 1) the state-action distribution from the dynamics data ``covers" the state-action pairs generated by $\pi^*$, which naturally includes stitching; 2) the support of $\mu_{goal}$ "covers" the goals $\pi^*$ would reach. The coverage is measured based on the generalization ability of $f$ and $g$; e.g., if $g(x_1)$ and $g(x_2)$ are similar for $x_1\neq x_2$, then $x_2$ is within the coverage so long as $x_1$ can be generated by $\mu_{goal}$.
>
> **3. “The motivation for why Algorithm 1 is expected to work is not given…In fact, the algorithm looks like it shouldn't work, since it gives zero rewards for all states in D_dyn potentially including goal states that were given rewards of 1 in D_goal. It is not clear why this reward scheme makes sense”**
>
> We present why our augmented MDP is equivalent to the original MDP in the sense of regret in Appendix B1, and add Remark 3.1 as an intuition. The core idea is that by introducing an extra “fictitious” action $a^+$, we create some fictitious transition from the potential goal states to the terminal state as future transitions after reaching the goal state. **For such potential goal states, the positive future reward will be propagated back to them with a high value by Bellman equation.** For example, if s0->...->g is a trajectory to some potential goal g, adding future transition g - > s+ with action a^+ and reward one, and labeling all previous trajectories with reward 0 will not affect the ranking of the learned value functions.
>
> **4. Presentation of the theoretical results**
>
> Thanks for your suggestions. We reorganized Section 3 accordingly (e.g., explanations, definitions). Please check the revised draft.
>
> ### Experiments
> **1. “All the experiments are in a single domain (the D4RL AntMaze)”**
>
> We chose AntMaze because it is a popular benchmark and it is easy to adapt its environments to construct offline CGO problems. (On the other hand, locomotion mujoco problems in D4RL are not goal oriented). . Besides the 3 variations, we also add goal prediction as an extra baseline and experiment with new perturbation settings (Section 4.4 and 4.5) when constructing the context-goal dataset, which helps to show the efficacy of the method in a better way.
>
> **2. 5 random seeds**
> Our empirical results are from over 100 evaluations from each random seed, and then averaged over 5 random seeds for each our empirical results are from over 100 evaluations from each random seed, and then averaged over 5 random seeds for each environment, which follows the convention of evaluation metrics in AntMaze.

---

> ### Author Response · Authors · 2023-11-21
> **Rebuttal (continued)**
>
> ### Limitations
> **1. “The proposed approach is only applicable to goal-conditioned tasks with rewards of 1 only at goal states and zero rewards otherwise.”**
>
> We agree such goal oriented problems do not cover the full spectrum of RL problems, but they are quite common in practice, especially given many real-world problems do not come with dense rewards naturally. That is why goal conditioned RL (GCRL) problems are widely studied in the literature. We choose the sparse reward (only 1 at goal states and 0 otherwise) setting as it is a realistic and commonly-used setting in goal conditional RL where the reward is sparse. It is studied in most of the GCRL papers [1]. We should note that this setting also covers problems generated by applying any reward shifting/scaling (e.g.,GCRL problem with -1 at each step and 0 at the goal) since they will induce the same policy ranking.
>
> [1] Goal-conditioned reinforcement learning: Problems and solutions. M Liu, M Zhu, W Zhang
>
> **2. "The paper makes very strong assumptions about the offline dataset. Mainly, they assume that the goals dataset D_goal contains all the goals the agent will encounter and that the trajectories dataset (D_{dyn}) contains trajectories leading to those goals.”**
>
> The assumption stated here is indeed a misinterpretation of our theory. We are sorry if our original manuscript’s writing is not accurate. To clarify, we only require 1) the distribution generating the dynamics dataset "covers" a feasible path to the target goal set and 2) the context-goal distribution "covers" some feasible goals. We remark that "covering'' here does not mean that such a path or a goal has to appear in the datasets or the data have to be generated only by goal-reaching policies (e.g., it suffices that data can be stitched together to form a goal-reaching path). Instead, it is a distribution-level coverage requirement; especially **we present coverage conditions based on the generalization ability of function approximators used in learning, which is weaker than requiring coverage conditions based on non-zero density**. We also add additional explanations in our assumptions and theorems in the draft.

---

### Author Response · Authors · 2023-11-21
**Overall Response**

We thank all reviewers for the thoughtful reviews and insightful questions. To make our paper easier to understand and more readable, we make major changes as below (marked in orange in the revised draft):
1.  We reorganize Section 3.3 with a more detailed explanation of our theoretical results. We notice there are some common misunderstandings of our assumptions in the reviews. We highlight that **our method only requires converge conditions at the distribution level which is measured in terms of the generalization ability of function approximators used in learning, rather than having the datasets to contain exactly feasible goals or paths to them**. Since our method is based on a reduction offline RL, it can also naturally use the base offline RL algorithm to **stitch trajectories together**.
2. We add extra remarks in Section 3.1 to explain why the proposed method should work in response to reviewers i3TD and Fsyr.
3. In response to the concerns about the missing goal prediction baseline method raised by reviewer Mndx, we add **extra experiments using the goal prediction baseline in Section 4.4 and 4.5**, where we show that our method also outperforms the goal prediction baseline.
4. In response to reviewer Fsyr and Mndx about the concerns that hindsight context labeling could be a naive solution, we also perturb the states when constructing the context-goal dataset **in Section 4.4 and 4.5 such that the goals in the context-goal dataset do not appear in the dynamics only dataset** (which means such hindsight context labeling is not possible) and show that our method still works well, which helps to show the efficacy of the method in a better way.
5. To better contextualize the contributions of our work in relation to the relevant literature,  We move related work back to the main text, at the end of the introduction.

---

### Meta-Review · Area_Chair_Tuu3 · 2023-12-05

**Metareview:**

Summary: The paper addresses the goal-conditioned offline reinforcement learning problem, where an agent learns to reach given goals based on offline training data. The authors propose the Simple Data Sharing (SDS) method, which involves augmenting the dataset with an absorbing action and state, adding rewards for goal states, and providing a theoretical analysis of the optimality of policies learned from the modified dataset. The experiments compare SDS with other offline RL methods, demonstrating its effectiveness in continuous control tasks, particularly in AntMaze environments.

Weaknesses: All experiments are confined to a single domain (D4RL AntMaze), limiting the generalizability of the proposed approach.
Experiment results are based on a small number of seeds (5), raising concerns about the robustness of empirical claims. Lack of a dedicated related works section in the main paper is highlighted as a limitation, and moving related works to the appendix is deemed excessive -- which authors addressed by moving back to the main paper.
Clarity of writing and notation are lacking. There is discrepancy between theory and practical solutions.

Strengths: The paper introduces an important problem and proposes a novel approach, SDS, for learning goal-conditioned policies from trajectory datasets with positive goal-state labels. The proposed algorithm is theoretically grounded, and the authors provide a theorem bounding the optimality of policies learned from the modified dataset. Empirical results demonstrate that SDS, when combined with offline RL, outperforms prior offline RL methods in AntMaze environments.

Suggestions: Changes to the experiment design to better showcase different properties of SDS and include more diverse benchmarks. Address discrepancy between theory and algorithm.

**Justification For Why Not Higher Score:**

All experiments are confined to a single domain (D4RL AntMaze), limiting the generalizability of the proposed approach.
Experiment results are based on a small number of seeds (5), raising concerns about the robustness of empirical claims.
Clarity of writing and notation are lacking. There is discrepancy between theory and practical solutions.

**Justification For Why Not Lower Score:**

N/A

---

### Decision · Program_Chairs · 2024-01-16

Reject